# A comprehensive survey and comparative analysis of time series data augmentation in medical wearable computing

Md Abid Hasan[1]*, Frédéric Li[1], Philip Gouverneur[2],
Artur Piet[2], Marcin Grzegorzek[1,2]‡

1 German Research Center for Artificial Intelligence (DFKI), Lübeck, Germany, 2 Institute of Medical Informatics, University of Lübeck, Lübeck, Germany

‡ This author supervise the whole project.
* abid1084@gmail.com

**Data availability statement:** The five datasets used in this study were acquired by third parties and are publicly available. They can be found as follows: OPPORTUNITY: Publicly available at

## Abstract

Recent advancements in hardware technology have spurred a surge in the popularity and ubiquity of wearable sensors, opening up new applications within the medical domain. This proliferation has resulted in a notable increase in the availability of Time Series (TS) data characterizing behavioral or physiological information from the patient, leading to initiatives toward leveraging machine learning and data analysis techniques. Nonetheless, the complexity and time required for collecting data remain significant hurdles, limiting dataset sizes and hindering the effectiveness of machine learning. Data Augmentation (DA) stands out as a prime solution, facilitating the generation of synthetic data to address challenges associated with acquiring medical data. DA has shown to consistently improve performances when images are involved. As a result, investigations have been carried out to check DA for TS, in particular for TS classification. However, the current state of DA in TS classification faces challenges, including methodological taxonomies restricted to the univariate case, insufficient direction to select suitable DA methods and a lack of conclusive evidence regarding the amount of synthetic data required to attain optimal outcomes. This paper conducts a comprehensive survey and experiments on DA techniques for TS and their application to TS classification. We propose an updated taxonomy spanning across three families of Time Series Data Augmentation (TSDA): Random Transformation (RT), Pattern Mixing (PM), and Generative Models (GM). Additionally, we empirically evaluate 12 TSDA methods across diverse datasets used in medical-related applications, including OPPORTUNITY and HAR for Human Activity Recognition, DEAP for emotion recognition, BioVid Heat Pain Database (BVDB), and PainMonit Database (PMDB) for pain recognition. Through comprehensive experimental analysis, we identify the most optimal DA techniques and provide recommendations for researchers regarding the generation of synthetic data to maximize outcomes from DA methods. Our findings show that despite their simplicity, DA methods of the RT family are the most consistent in increasing performances compared to not using any augmentation.

https://archive.ics.uci.edu/dataset/226/opportunity+activity+recognition HAR : Publicly available at https://archive.ics.uci.edu/dataset/240/human+activity+recognition+using+smartphones DEAP: Access granted after signing an EULA https://www.eecs.qmul.ac.uk/mmv/datasets/deap/ Biovid: Access granted after signing an EULA https://www.nit.ovgu.de/BioVid.html PainMonit: Publicly available at https://figshare.com/articles/dataset/The_PainMonit_Database_An_Experimental_and_Clinical_Physiological_Signal_Dataset_for_Automated_Pain_Recognition/26965159.

**Funding:** This study was supported in part by the Deutscher Akademischer Austauschdienst (Award No. 91831212), German Research Center for Artificial Intelligence (DFKI), and within the grant SAM-SMART "Security Assistance Manager for the Smart Home" (Bundesministerium für Bildung und Forschung, Grant No. 16KISA074). The funders had no role in study design, data collection and analysis, decision to publish, or preparation of the manuscript.

**Competing interests:** The authors have declared that no competing interests exist.

**Abbreviations: ACC**, Accelerometer; **ADL**, Activities of Daily Living; **AE**, Autoencoder; **AI**, Artificial Intelligence; **ANN**, Artificial Neural Network; **BLC**, Binary Label Classification; **BLSTMCNN**, Bidirectional LSTM-CNN; **BPS**, Behavioural Pain Scale; **BVDB**, BioVid Heat Pain Database; **BVP**, Blood Volume Pulse; **BVPD**, BioVid Heat Pain Database; **cGAN**, Conditional Generative Adversarial Networks; **CNN**, Convolutional Neural Network; **D**, Discriminator; **DA**, Data Augmentation; **DeepConv2D**, Deep Convolutional 2D Neural Networks; **DeepConvLSTM**, Deep Convolutional and LSTM Recurrent Neural Networks; **DGW**, Discriminative Guided Warping; **DL**, Deep Learning; **DNN**, Deep Neural Network; **DTW**, Dynamic Time Warping; **EDA**, Electrodermal Activity; **EDA-E4**, Electrodermal Activity from the Empatica E4; **EDA-RB**, Electrodermal Activity from the respiBAN Professional; **ECG**, Electrocardiogram; **EEG**, Electroencephalogram; **EMDA**, Equalized Mixture Data Augmentation; **EMG**, Electromyograms; **EOG**, Electrooculogram; **G**, Generator; **GAN**, Generative Adversarial Networks; **GM**, Generative Models; **GSR**, Galvanic Skin Response; **HAR**, Human Activity Recognition; **HR**, Heart Rate; **IBI**, Inter-Beats-Interval;

# 1 Introduction

## 1.1 Background

Wearable sensors play a pivotal role in healthcare, offering continuous monitoring of vital signs and yielding valuable insights into individuals' health status. The data derived from these sensors hold significant potential for clinical decision-making, facilitating personalized interventions, and advancing scientific understanding of physiological processes. The increasing availability of such data has opened up numerous opportunities to leverage machine learning techniques to automate high-level medical tasks. However, the challenging nature of data acquisition and labeling results in datasets being both scarce and small. In recent times, there has been rising interest in the applications of machine learning to analyze Time Series (TS) data to solve tasks such as classification [1], forecasting [2], and anomaly detection [3], especially in a supervised learning context. Specifically, the utilization of deep learning for analyzing TS data obtained from wearable sensors has witnessed substantial growth in the fields of wearable-based human activity recognition [4], analysis of pain [5,6] and emotion [7,8].

## 1.2 Current challenges

The success of supervised learning depends on a large number of training examples, each of which must be labeled, in order to enhance the model performances and generalization capabilities [9–11]. The primary obstacle to obtaining a robust ML model lies in obtaining sufficient data to train it; this challenge is particularly pronounced when dealing with medical data. A natural way to address this issue is to generate synthetic data using Data Augmentation (DA) methods. DA aims to enrich a training dataset by generating synthetic data using various approaches such as transformations, pattern mixing, and pattern generation [12]. In the context of TS classification, DA serves two primary purposes: (**I**) enhancing the model generalization capacity to reduce overfitting, and (**II**) addressing class imbalances within the dataset. More specifically, DA has been shown to improve model adaptability on unknown data, especially when image modalities are used [13,14]. For instance, the pioneering deep Convolutional 28 Neural Network (CNN) AlexNet [10] was trained with augmented data obtained from cropping, mirroring, and color augmentation, achieving a record benchmark in the ImageNet Large Scale Visual Recognition Challenge (ILSVRC) dataset [15]. Similarly, other notable examples such as the Visual Geometry Group (VGG) network [16], Residual Networks (ResNet) [17], DenseNet [18] and inception networks [19] all have integrated various augmentation techniques such as jittering, slicing, drifting, rotation, and color augmentation into their original proposals. In 2014, Goodfellow et al. proposed the Generative Adversarial Networks (GAN), an influential deep-learning-based approach that generates new data based on an adversarial learning scheme instead of property transformation [20]. GAN are widely used and adopted algorithm in DA for image datasets [21,22]. However, while employing DA in image-based training processes is a widespread practice, especially when Neural 40 Networks (NN) are involved, it has not yet become a standard procedure for time series recognition. The diverse characteristics found in TS data, including trends, seasonality, periodicity, and irregular patterns, make it challenging to apply DA techniques universally across TS dataset.

TSDA presents several challenges that are distinct from those encountered in other data types, such as images to text. These challenges stem from the inherent complexity of TS data, more specifically regarding its sequential nature, temporal dependencies, increased difficulty to interpret, and the need to preserve the data integrity throughout the augmentation process.

**LOSO**, Leave-One-Subject-Out; **LSTM**, Long Short-Term Memory; **ML**, Machine Learning; **MLC**, Multi-Label Classification; **MLP**, Multi-Layer Perceptron; **MW**, Magnitude Warping; **NN**, Neural Networks; **PM**, Pattern Mixing; **PMDB**, PainMonit Database; **PRM**, Permutation; **RGW**, Random Guided Warping; **RNN**, Recurrent Neural Network; **RT**, Random Transformation; **SCL**, Skin Conductance Level; **sEMG**, Surface Electromyogram; **SMOTE**, Synthetic Minority Oversampling Technique; **SPAWNER**, SuboPtimAl Warped Time Series GeNEratoR; **TS**, Time Series; **TSDA**, Time Series Data Augmentation; **TSC**, Time Series Classification; **TW**, Time Warping; **UCR**, University of California Riverside; **VAE**, Variational Autoencoder; **WW**, Window Warping

A first significant challenge of TSDA is the overall scarcity of TS data [12,23]. Unlike image or text data which are abundant, TS data can be costly to acquire both in time and resources, in particular in medical-related fields. As a result, the available TS datasets are usual small, and fragmented in the sense that merging datasets containing data acquired from different sensors or devices is a non-trivial process. This situation can make the application of generative approaches that rely on large amount of training data, such as GANs, quite complex in practice. Furthermore, labeled TS datasets frequently exhibit class imbalance, where certain events or behaviours to recognise are underrepresented. This combined with the data scarcity increases the difficulty for machine learning models to generate meaningful samples for the least represented classes. As a result, existing augmentation techniques often fail to generate sufficiently diverse synthetic data or introduce unrealistic patterns that do not accurately reflect the temporal structure of the original data [12,24,25]. Label integrity is a second concern in TSDA. Unlike image data, where transformations like flipping, rotation, and scaling do not alter the semantic meaning of the data, TS data is highly sensitive to such transformations. Even minor alternations can change the underlying patterns that define a class level. This necessitates the development of specialised augmentation techniques that ensure the augmented sample retain the same label as the original sample it is coming from, a challenge that remains mostly unresolved for now [26]. A last major challenge lies in the difficulty of assessing the quality of the generated TS samples. In image data augmentation, the quality of the generated data can be relatively easily evaluated through visual inspection, allowing humans to verify whether the augmented images maintain the characteristics of the original ones and retain the same semantic meaning. However in the context of TS data, direct visual validation is usually not trivial, especially for complex physiological data that is for instance acquired in medical-related applications. Small distortions in the augmented TS that cannot be easily detected can significantly affect the patterns and trends within the data, potentially leading to incorrect predictions when models are trained on these augmented datasets. Current methods for evaluating the quality of synthetic time series data are limited, making it challenging to ensure that the augmented data follows a distribution similar to the one of the original data [25].

Investigations to address these challenges are currently still on-going in the literature related to TSDA. As a result, most of the commonly used TSDA approaches either bypass data scarcity issues by not requiring a data learning-based process, or hinge on the assumption that small modifications to the original TS preserve its class label and semantic content.

## 1.3 Research gaps

Several investigations for TSDA have been made in the past literature. Preliminary work was undertaken by Fawaz et al. [1], in which comprehensive experiments were conducted utilizing the benchmark TS classification dataset from the University of California Riverside (UCR) [27] to assess the effectiveness of augmenting the training set. The experiments conducted in the paper are however limited to univariate data only, and the generalizability of their findings to other TS classification datasets is not explored. Iglesias et al. [25] provided a systematic review of the current state-of-the-art DA techniques for TS. Nevertheless, their study did not carry out any comparative experiments; instead, it solely surveyed existing methods documented in the literature. Wen et al. [12] proposed their own TSDA taxonomy and expanded the scope of their experiments to encompass TS forecasting and anomaly detection. However, their investigation relied on a single and relatively niche dataset derived from the Alibaba Cloud monitoring system [12]. Iwana et al. [24] proposed a complete taxonomy of TSDA techniques that classifies them into three main families: Random

Transformation (RT), Pattern Mixing (PM), and Generative Model (GM), themselves divided into time, magnitude, and frequency domains. They additionally empirically investigated various TSDA methods using NN, with a specific emphasis on TS classification and excluding considerations of all the frequency domain algorithms and advanced generative models. However, this comparative study omitted the inclusion of GM methods, leading to an incomplete evaluation of effective methodologies for TSDA. Gao et al. [28] presents a survey of existing DA methods for TS classification. The study however does not examine DA methods for multivariate TS datasets. Additionally, it lacks recommendations or guidelines for researchers regarding the optimal extent of augmentation for training data.

Although all of the aforementioned studies provide valuable insights regarding DA techniques, there is a noticeable lack of a comprehensive perspective that uses both univariate and multivariate TS datasets, includes all types of DA approaches (RT, PM and GM) in an experimental comparative study and covers diverse application fields.

To be more specific, the research gap from the previous study can be highlighted as follows:

- Multivariate TS dataset: None of the reviewed studies, thoroughly investigate TSDA techniques in the context of multivariate time series data. There is a significant gap in understanding how these methods perform when applied to datasets with multiple correlated time series.
- Comprehensive comparative analysis: While some studies reviewed various DA techniques, they either did not conduct empirical comparisons [25], or did not include all relevant methodologies, such as advanced generative models [24]. This leaves a gap in knowledge about the relative effectiveness of different DA approaches across diverse datasets and application domains.
- Recommendations for optimal augmentation: None of the existing surveys provide practical guidelines or recommendations for the extent of data augmentation, which is essential for optimizing the performance of TSDA methods in real-world applications.
- Experimentation on medical datasets: Most past studies have been carried out on generalist TS data repositories, such as the UCR Univariate data archive [12,24,27]). Our study offers a more specific overview of TSDA on medical datasets, and tests their generalizability and robustness across various dataset related to medical wearable computing.

This paper aims to bridge this gap by offering a thorough analysis of the current landscape of TSDA for both univariate and multivariate TS data based on the taxonomy proposed by Iwana et al. [24]. We hope our study can serve as a guide for future research directions in this rapidly evolving field.

## 1.4 Main contributions

The objective of this research is to address the aforementioned gaps by summarizing and explaining the prevalent TSDA approaches to TS classification tasks. DA methods from selected families are experimentally compared on several TS benchmark datasets to generate some practical advice for the application of TSDA.

More specifically, the main contributions of this research are summarized as follows:

1. We describe a taxonomy for time series classification ensuring applicability across diverse datasets, and overcoming the limitations found in previous studies. We also provide an extensive overview and description of TSDA techniques for both univariate

and multivariate datasets, covering their characteristics, and recommendations for use with various datasets.

2. We present a comprehensive comparative study of TSDA for classification tasks across diverse wearable computing applications (human activity recognition, pain, and emotion recognition) and properties (balanced, imbalanced, univariate, and multivariate).

3. We include GM DA techniques in our experimental analysis. To the best of our knowledge, no previous research work has presented the GM in a TSDA comparative study for multivariate datasets.

4. We propose an effective strategy for generating a certain volume of synthetic data from DA techniques to enhance the data quality and model performance.

5. We present future directions and suggestions, unveiling effective algorithms tailored for various time series data types.

Additionally, we share our implementation of the tested TSDA techniques at the following repository: https://github.com/hasanmdabid/DA_Project.git (last accessed on 08.11.2024).

The rest of the article is structured as follows: Sect 2 describes the recent literature related to the topic of TSDA. hyperref[sec:methods]Sect 3 provides a comprehensive description of selected methods. Sect 4 presents the datasets and models selected for the study. Sect 5 presents the experimental results. Sect 7 provides a detailed discussion of our experimental results and findings. Finally, Sect 8 concludes this work and provides suggestions for future studies.

## 2 Related works

In recent years, an increasing number of articles on the topic of DA have been published. Most of them are focused on areas such as computer vision [13,29], and natural language processing [30,31] due to the large amount of data available for such applications. However, less attention has been paid to TSDA in the literature. We organize the TSDA related work from the literature in four categories: RT augmentation, PM augmentation, GM augmentation, and TSDA surveys.

**RT**: The majority of augmentations applied to TS data involve RT. Jittering for TS has been most commonly applied to sensor data; for instance, Um et al. [32] applied multiple RT-based DAs, including jittering for Parkinson's disease classification based on wearable sensor data. However, they were unable to enhance the performances of their model with DA, as all of their tested approaches decreased the accuracy of their baseline CNN model. In the context of TS classification with NNs, rotation augmentation has been observed to either have no impact or to be detrimental [32,33]. On the contrary, Hasan et al. [34] demonstrated rotation DA improves the $F1$-Score of human activity recognition classification. In the same study, they reported that Time Warping (TW) performs best to improve the human activity recognition accuracy among other RT algorithms.

Even though the majority of existing methods concentrate on the time domain, some research works examine DA for TS from a frequency domain perspective. Frequency transformations modify the frequency components of the TS data and are widely used for applications where they carry meaningful information, such as in audio or vibration analysis. Frequency warping is a widely used DA technique in audio and speech recognition [35,36]. In order to prevent overfitting and enable more robust classification, [37] proposes a technique called Random Noise Boxes for DA of spectrograms. Essentially, each spectrogram is multiplied by a predetermined number of identical spectrograms, and parts of the resulting spectrograms are replaced by boxes with random pixels.

**PM**: Mixing patterns in the time domain operates on the timestamp level, keeping its original timing structure while creating new instances. Forestier et al. [38] proposed to use the Dynamic Time Warping (DTW) algorithm to mix a sample pattern with a reference pattern of the same intra-class to create synthetic examples. This method is particularly useful when there are not many examples available. Similarly, Iwana et al. [39] proposed a new method named guided time-warping that uses the dynamic alignment function of DTW to warp the elements of a sample pattern to the elements of a reference pattern. It can be classified into two types: Random Guided Warping (RGW), where reference patterns are selected randomly within the same class, and Discriminative Guided Warping (DGW), a method where the reference patterns are selected using class label information. The pattern chosen by the discriminator is called a discriminative teacher and it is defined by finding the sample within a bootstrap set with the maximal distance between the patterns of the same class and patterns of a different class.

The main goal of frequency-domain PM techniques is to manipulate the frequency components of TS data. These techniques are essential when the frequency content of the data is important since they allow for the creation of new instances while maintaining the frequency structure of the original data. A comparative study focusing on the PM approaches to TSDA was reported in [40]. The authors of Nanni et al. [41] describe a spectrogram-based method called Equalized Mixture Data Augmentation (EMDA), which involves creating a new spectrogram by averaging two randomly chosen spectrograms from the same class and adding perturbations such as equalizer functions and time delays. The strategy increases the robustness and diversity of audio signals, strengthening classification performance. Alternatively, Cui et al. [36] promotes the use of stochastic feature mapping, a label-preserving transformation that statistically converts speech data from one speaker into another within a given feature space. Since the frequency domain is used, DAs in this field are typically used for sound recognition; for example, EMDA has been used for animal audio classification [41], acoustic event detection [36].

**GM**: A statistical GM for DA is a probabilistic framework used to generate synthetic data samples resembling the distribution of the original dataset. This model is particularly used for prediction purposes. Smyl et al. [42] used a long short-term memory (LSTM) network for the forecasting task, where they augmented the data in the preprocessing step by Local and Global Trend model. Cao et al. [43] describe a mixture of the Gaussian Trees model intended for DA in situations involving imbalanced and multimodal TS classification. They highlight the model reduces storage complexity and outperforms other oversampling techniques, particularly in datasets with multimodal minority class distributions.

NNs-based generative models have gained popularity recently. In 1987, Yann Lecun first introduced the idea of Autoencoder (AE) in his thesis and mentioned that these networks can be used for the generation of data [44]. An encoder and a decoder make up the AE network. The encoder is responsible for reducing the dimensionality of the input data to a latent space, and the decoder reconstructs the input information from this latent representation. The latent space is a lower-dimensional manifold of the input data. Next, synthetic data is produced by interpolating the values of the latent space and decoding them. However, this process of interpolating the latent space does not produce entirely new values; rather, it merely mixes the features of the learned probability distribution. To address this problem, Kingma and Welling proposed the Variational Autoencoder (VAE) in Kingma et al. [45]. In VAE, instead of generating the latent space point, VAE encodes the input information in a probability distribution. It then samples a point from this distribution, which is subsequently decoded to generate augmented data. Alawneh et. al enhanced human activity recognition classification using deep

learning and augmented data by VAE [46]. Moreover, improvements in the overall accuracy of TS classification on different datasets (e.g. seismic data, UCR dataset) are reported in the related literature [47–49].

Researchers have proposed a modified version of GAN in the form of conditional Generative Adversarial Networks (cGAN) to take class labels into account during the generation process. To classify human emotion, Chen. et. al proposed EmotionGAN that uses 1D CNN as generator [50]. To generate the HAR signals, ActivityGAN is proposed by Li et al. [51]. The architecture comprises a Generator (G) which is a stack of 1D-convolution and transposed convolution (1D-transposed convolution) layers, and a Discriminator (D) model which employs two-dimensional convolution networks (2D-convolution). Hasan et al. [34] applied cGAN (using 2D CNN for both G and D architectures) to balance the human activities dataset by generating examples from the least represented class(es) and improve the accuracy of deep neural networks. A hybrid method, called HybridGAN, is proposed by Zhu et al. [52] and combines the LSTM cell in the D and 1D convolutional layers used in the G network. Later, Zhao et al. [53] used a bidirectional LSTM-CNN hybrid model and claimed that the modified hybrid model is better than the LSTM-CNN model. An effort to combine both VAE and GAN for TS prediction is reported in [49].

**TSDA surveys**: Preliminary work on DA was undertaken by Fawaz et al. [1], in which comprehensive experiments were conducted utilizing the benchmark TS classification dataset from the UCR [27] to assess the effectiveness of augmenting the training set. The experiments conducted in the paper are however limited to univariate data only, and the generalizability of their findings to other TS classification datasets is not thoroughly explored. Iglesias et al. [25] provide a systematic review of the current state-of-the-art DA techniques for TS. Nevertheless, their study did not carry out any comparative experiments; instead, it solely surveyed existing methods documented in the literature. Iwana et al. [24] empirically investigated various TSDA methods using NNs, with a specific emphasis on TS classification and excluding considerations of all the frequency domain algorithms and advanced generative models. Their findings showed that the Window warping (WW) algorithm from RT domain stands out as the most effective DA algorithm specifically tailored for univariate TS datasets. Subsequently, Wen et al. [12] expanded the scope of this study to encompass TS forecasting and anomaly detection; however, their investigation relied on a relatively niche dataset derived from the Alibaba Cloud monitoring system [12]. One work closely related to ours was proposed by Gao et al. [28] which presents a survey of existing DA methods for TS classification. The study however does not examine DA methods for multivariate TS datasets. Additionally, it lacks recommendations or guidelines for researchers regarding the optimal extent of augmentation for training data.

Although the aforementioned studies provide valuable insights regarding DA techniques, there is a noticeable lack of a comprehensive perspective that encompasses recent advancements and diverse application fields. This paper aims to bridge this gap by thoroughly analyzing the current landscape of TSDA, guiding future research directions in this rapidly evolving field. Ours is the first study to apply, compare, and recommend effective DA for multivariate and multi-domain TS, more specifically in several medical-related application fields.

## 3 Methodology

The subsequent sections provide a thorough discussion of the TSDA methods selected for this study, highlighting their respective advantages, and potential applications. This

comprehensive analysis accomplishes two goals: it helps users choose appropriate DA procedures and stimulates the creation of novel techniques for further research endeavors.

## 3.1 Proposed taxonomy

In the past, researchers have suggested various taxonomies for data augmentation (DA) algorithms; however, these classifications do not always accommodate the specific needs of different time series (TS) formats. More specifically, a thorough review of the existing literature [24,28] showed that the previously existing frameworks were not necessarily addressing the nuances of applying DA methods to both univariate and multivariate time series, to datasets with varying periodicities and structures, or with different class distributions in the case of labeled datasets (i.e. balanced or unbalanced datasets). To fill this gap, we propose a taxonomy of DA approaches that can be applied to both of these types of time series data. Our categorization builds on and refines these earlier research efforts by focusing specifically on techniques that are widely used in the field. With this objective in mind, we decided to build our taxonomy upon the one proposed by Iwana et al. [24], as we considered it to be the most comprehensive among the ones that we found, and the easiest to use for a practical application.

Fig 1 illustrates the DA techniques examined in this study. To ensure broad applicability, we selected TSDA methods that can be used across different types of time series datasets. We specifically chose to exclude frequency domain-based augmentation methodologies, as these techniques are generally suitable only for periodic time series [24]. Additionally, while statistical generative models are frequently employed in forecasting tasks [42,54–56], our focus is on time series classification. Therefore, we excluded statistical models from our study to concentrate on methods that are more directly relevant to classification tasks. Moreover, Synthetic Minority Oversampling technique (SMOTE) was omitted based on findings by Hasan et al. [34], which indicated that SMOTE struggles to generate meaningful time series data when applied to large training datasets. All the DA methods selected in our taxonomy were applied in an experimental study involving five labeled TS datasets related to medical applications, and comprising diverse characteristics in terms of data type (univariate or multivariate) and class distribution (balanced or unbalanced).

## 3.2 Random transformation

The underlying idea of RT-based methods is to apply various transformations to the original TS data. The goal is to preserve their basic structure while incorporating subtle variations that increase the robustness of the model. These methods are known for their simplicity and ease of use. They usually involve simple mathematical operations, making them applicable to various data types and domains, as illustrated in Fig 2. Transformations can be applied to three different aspects, leading to a categorization into three subdomains.

**3.2.1 Magnitude domain RT** Magnitude transformations modify the amplitude or value of TS data while the time steps are kept constant, which is particularly useful in applications like audio analysis and signal processing where the magnitude of the data provides meaningful information. In our study, we describe jittering, rotation, scaling, and Magnitude Warping (MW) from the family of magnitude domain RT. To explain the methods, we denote a TS as $\mathbf{X} \in \mathbb{R}^{T \times S}$ where $T \in \mathbb{N}^*$ is its length along the time dimension and $S \in \mathbb{N}^*$ is its dimensionality. $\mathbf{X}$ is an univariate TS when $S = 1$, and multivariate when $S > 1$.

$$\mathbf{X} = (x_{ij})_{\substack{1 \le i \le T \\ 1 \le j \le S}} \tag{1}$$

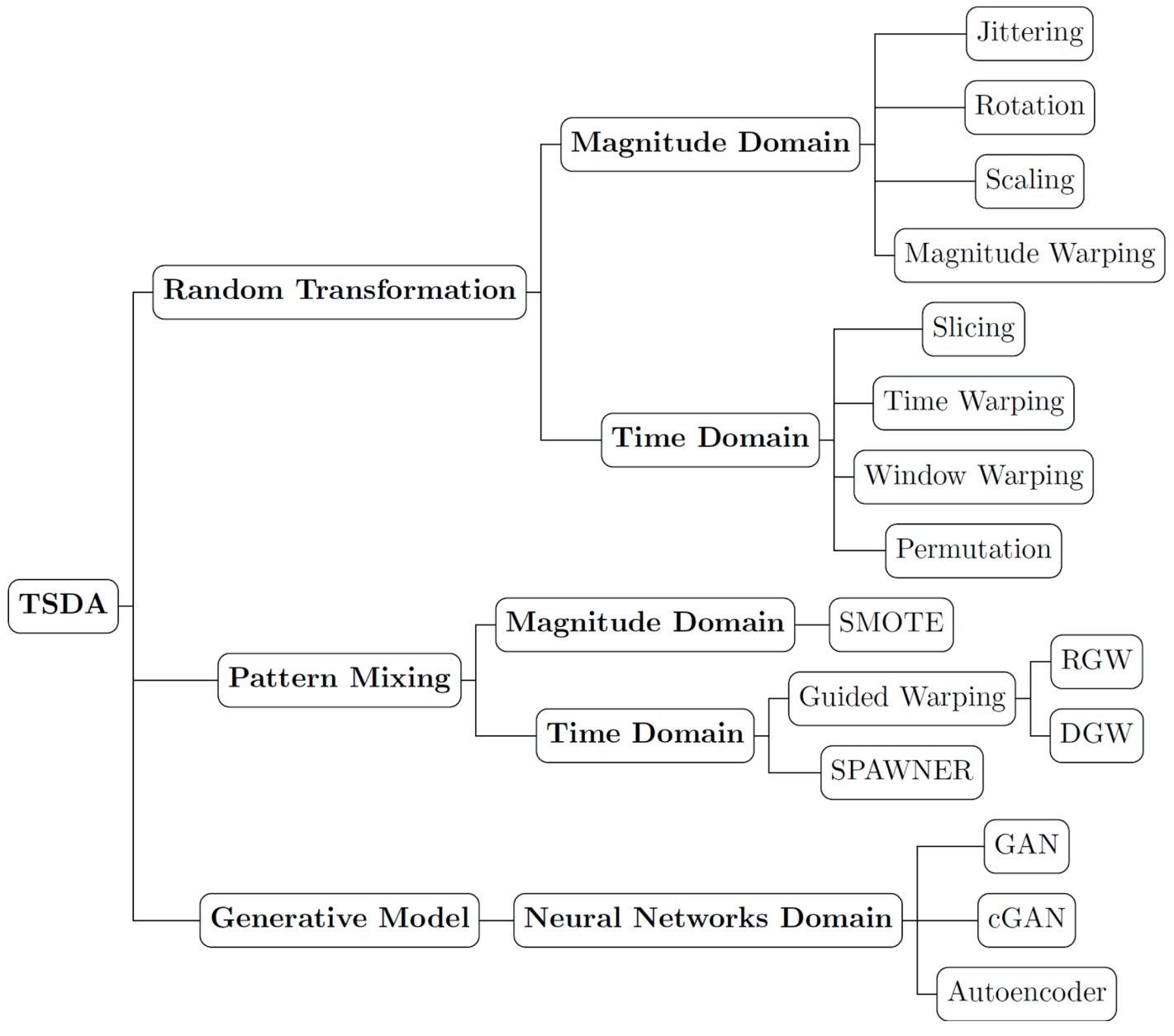

**Fig 1. Taxonomy of TSDA methods.**

We identify the most commonly used magnitude domain RT DA techniques as jittering, rotation, scaling and MW, and subsequently describe them. Additionally, an example of a time series example augmented with each technique is presented in Fig 2.

**Jittering** consists in adding Gaussian noise $\mathbf{\Omega}(t) \in \mathbb{R}^{T \times S}$ to each time step of the original signal to obtain a new TS $\mathbb{X}' \in \mathbb{R}^{T \times S}$ to augment the training set, as shown in Eq 2:

$$\mathbf{X}' = \mathbf{X} + \mathbf{\Omega} = \begin{pmatrix} x_{11} + \omega_{11} & \cdots & x_{1S} + \omega_{1S} \\ \vdots & \ddots & \vdots \\ x_{T1} + \omega_{T1} & \cdots & x_{TS} + \omega_{TS} \end{pmatrix} \tag{2}$$

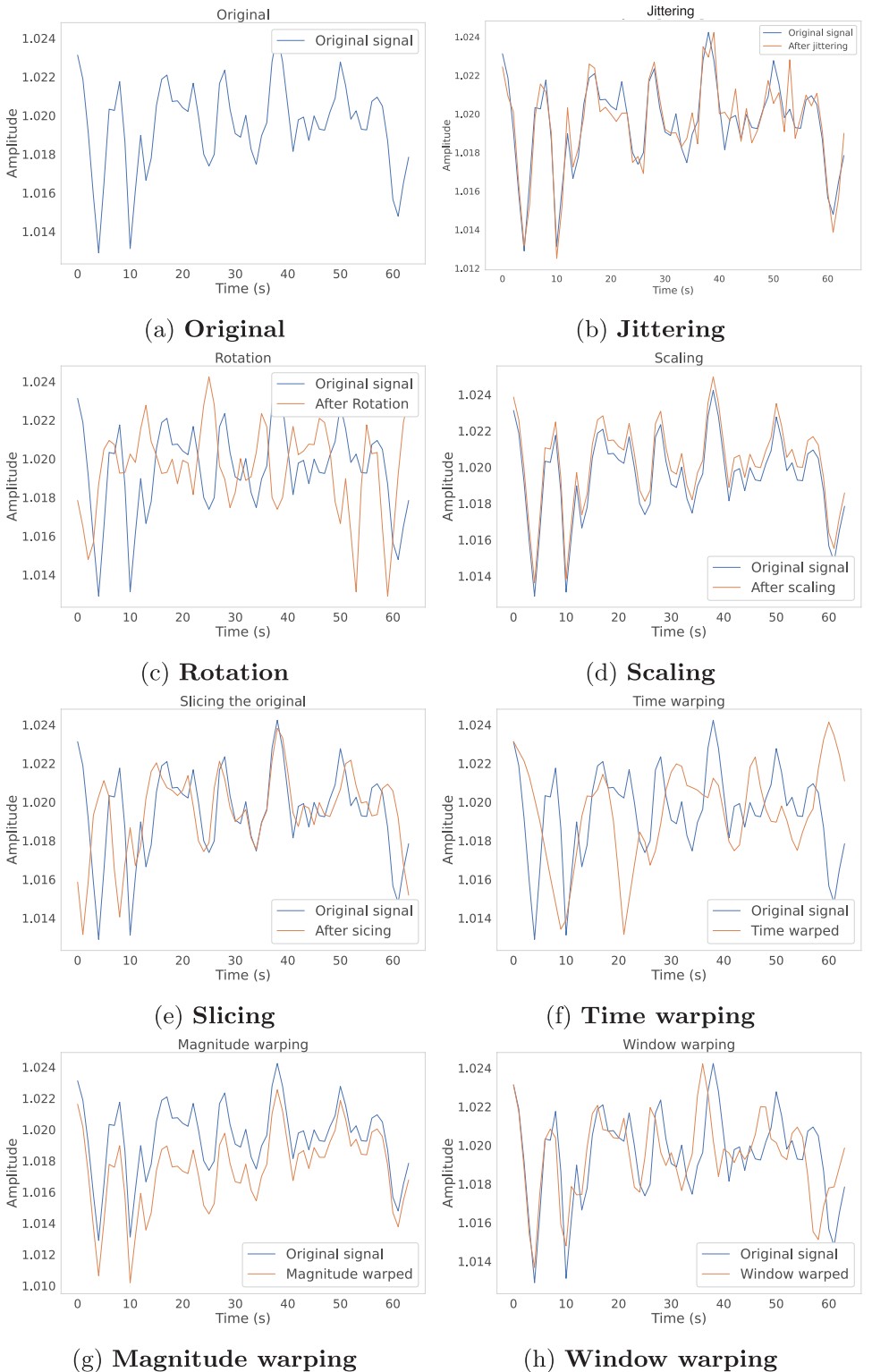

**Fig 2. Example instances of RT methods applied to the sensor channels of the HAR dataset.** Fig 2b and 2c illustrate the effect of adding Gaussian noise and the rotation of the original signal. The original signal is scaled down by the scaling parameter $\alpha = 0.1$ in Fig 2d. An illustration of 90% sliced signal is presented in Fig 2e. Fig 2f, 2g, and 2h depicts the effects of different warping algorithm for knot $I = 4$, $\mu = 1$ and $\sigma = 0.2$. In Fig 2h the original signal is stretched along the time domain by a fold of 0.5 (left) and 2 (right).

with

$$\mathbf{\Omega} = (\omega_{ij})_{\substack{1 \leq i \leq T \\ 1 \leq j \leq S}} \quad \text{so that} \quad \omega_{ij} \sim \mathcal{N}(\mu, \sigma^2)$$

where $\mu, \sigma \in \mathbb{R}$ are parameters that designate the mean and standard deviation of the noise.

**Rotation** (or flipping), as defined by Iwana et al. [24], is an element-wise scaling of the time series elements performed by multiplying each of them by a factor $cos(\theta)$ with $\theta$ drawn according to a uniform distribution on $[0, 2\pi[$. It should be noted that this operation does not perform an actual mathematical rotation. The rotation augmentation follows Eq 3:

$$\mathbf{X}' = \mathbf{X} \odot \mathbf{R} \tag{3}$$

where $\odot$ designates the element-wise product, and $\mathbf{R} \in \mathbb{R}^{T \times S}$ is the rotation matrix defined as:

$$\mathbf{R} = (cos\theta_{ij})_{\substack{1 \leq i \leq T \\ 1 \leq j \leq S}} = \begin{pmatrix} cos\theta_{11} & \cdots & cos\theta_{1S} \\ \vdots & \ddots & \vdots \\ cos\theta_{T1} & \cdots & cos\theta_{TS} \end{pmatrix}$$

with $\forall i \in \{1, ..., T\}, \quad \forall j \in \{1, ..., S\}, \quad \theta_{ij} \sim U(0, 2\pi)$.

**Scaling**, also sometimes referred to as drifting, changes the magnitude of a TS by using a random multiplicative scaling value ($\alpha \in \mathbb{R}^*$) without altering the step sequences, as shown in Eq 4. The authors applied the scaling with other DA in [32–34].

$$\mathbf{X}' = \alpha \mathbf{X} \tag{4}$$

where $\alpha \in \mathbb{R}^*$.

**MW** alters the magnitude of individual samples by convolving the data window with a smoothly varying curve [24], as shown in Eq 5. The concept of magnitude warping revolves around the notion that minor fluctuations within the data can be introduced by increasing or decreasing random regions within the TS.

$$\mathbf{X}' = (\alpha_i \mathbf{x}^{(i)})^{\mathsf{T}}_{1 \leq i \leq T} \tag{5}$$

where $\mathbf{x}^{(i)} \in \mathbb{R}^S$ designates the $i^{th}$ row of $\mathbf{X}$, and $(\alpha_k)_{1 \leq k \leq T}$ represents a sequence of multipliers generated by the values of a cubic spline denoted as $S(u)$, with knots located at $u = (u_i)_{1 \leq i \leq I}$ where $I \in \mathbb{N}^*$ is the chosen number of knots. Each knot $u_i$ is sampled from a distribution of $\mathcal{N}(1, \sigma^2)$. Both the number of knots $I$ and the standard deviation $\sigma$ serve as hyperparameters in this context [24].

**3.2.2 Time domain RT** Time domain transformations closely resemble magnitude domain transformations, with the key distinction being that the transformation occurs along the time axis. Under the time domain of random transformation based DA, We cover slicing, TW, window warping, and permutation in our study.

**Slicing** (also sometimes referred to as window slicing [57]) involves selecting and retaining a segment or subset of the original TS while discarding the remaining portion. This technique is akin to its counterpart in image DA (cropping) but is applied along the temporal axis of the TS data, as illustrated by Eq 6:

$$\mathbf{X}' = (x_\zeta, ..., x_{\zeta+\omega}) \tag{6}$$

where $w$ is the size of the segmented window and $\zeta$ is a random integer such that $1 \leq \zeta \leq T - w$. It should be noted that $\mathbf{X}' \in \mathbb{R}^w$ with $w < T$. To match the length of the original TS, the sliced segment is usually stretched and resampled to fit the original TS timestamps using linear interpolation.

**Permutation** refers to the process of generating new data patterns by shuffling time slices of data. This method, initially proposed in [32], involves the rearrangement of data within a fixed slice window. In recent studies, permutation has been explored with variable windows, as discussed in [58]. However, a notable limitation of applying permutation lies in its inability to preserve time dependencies, potentially resulting in the generation of invalid samples. The algorithmic representation of the permutation process is shown in Eq 7:

$$\mathbf{X}' = (x_{\phi(1)}, x_{\phi(2)}, ..., x_{\phi(T)}) \tag{7}$$

where $\phi$ is a bijection of $\{1, 2, ....., T\}$ to itself.

**TW** is a DA approach that alters the temporal dimension curve. More specifically, it substitutes the time scale of the original TS with a new one, as illustrated in Eq 8:

$$\mathbf{X}' = (x_{\tau_{(1)}}, x_{\tau_{(2)}}, ..., x_{\tau_{(T)}}) \tag{8}$$

where $\tau$ defines a warping function that warps the time steps based on a smooth curve. The smooth curve function is constructed by utilizing a cubic spline $S$ between multiple knots $(u_i)_{1 \leq i \leq I}$, with $I \in \mathbb{N}^*$ being the chosen number of knots. The heights at the knots $S(u_i)$ are sampled from a normal distribution with mean $\mu$ and variance $\sigma^2$. This algorithm has been utilized in several papers, including [32,34,59,60].

**WW** is a variation of TW that was introduced by Le Guennec et al. [57]. In WW, two random non-overlapping subsequences with the same length are extracted from the TS to augment, and then respectively stretched by a factor $\beta = 2$ and contracted by a factor $\beta = 1/2$. Such a modification ensures that the augmented data sample has the same size as the original one it came from.

## 3.3 Pattern Mixing

Pattern-Mixing-based methods focus on identifying particular patterns or structures present in the TS data. Combining multiple identified patterns to create new ones is known as pattern mixing DA. These patterns can be simple or complex, local or global, and are frequently identified and used through ML models. An example of augmentation using PM techniques is shown in Fig 3.

**3.3.1 Magnitude domain PM** The most direct application of pattern mixing is to linearly combine the samples at each time step. This is the idea behind magnitude-domain mixing. At first, the difference between a sample and its nearest neighbor is calculated from the minority class dataset. By computing a weighted average of the two samples, a new sample is generated by the interpolation. SMOTE is a prominent example of an interpolation approach that was proposed to address imbalanced datasets [61]. SMOTE augments a sample $\mathbf{X}$ by first looking for its nearest neighbour $\mathbf{X_{nn}}$, then scaling the difference by a random factor $\lambda \in ]0, 1[$, and finally adding this difference to the original sample to produce an augmented example $\mathbf{X}'$ as shown in Eq 9. In practice, SMOTE is usually applied to only the examples of the minority class(es) to rebalance the dataset.

$$\mathbf{X}' = \mathbf{X} + \lambda(\mathbf{X_{nn}} - \mathbf{X}) \tag{9}$$

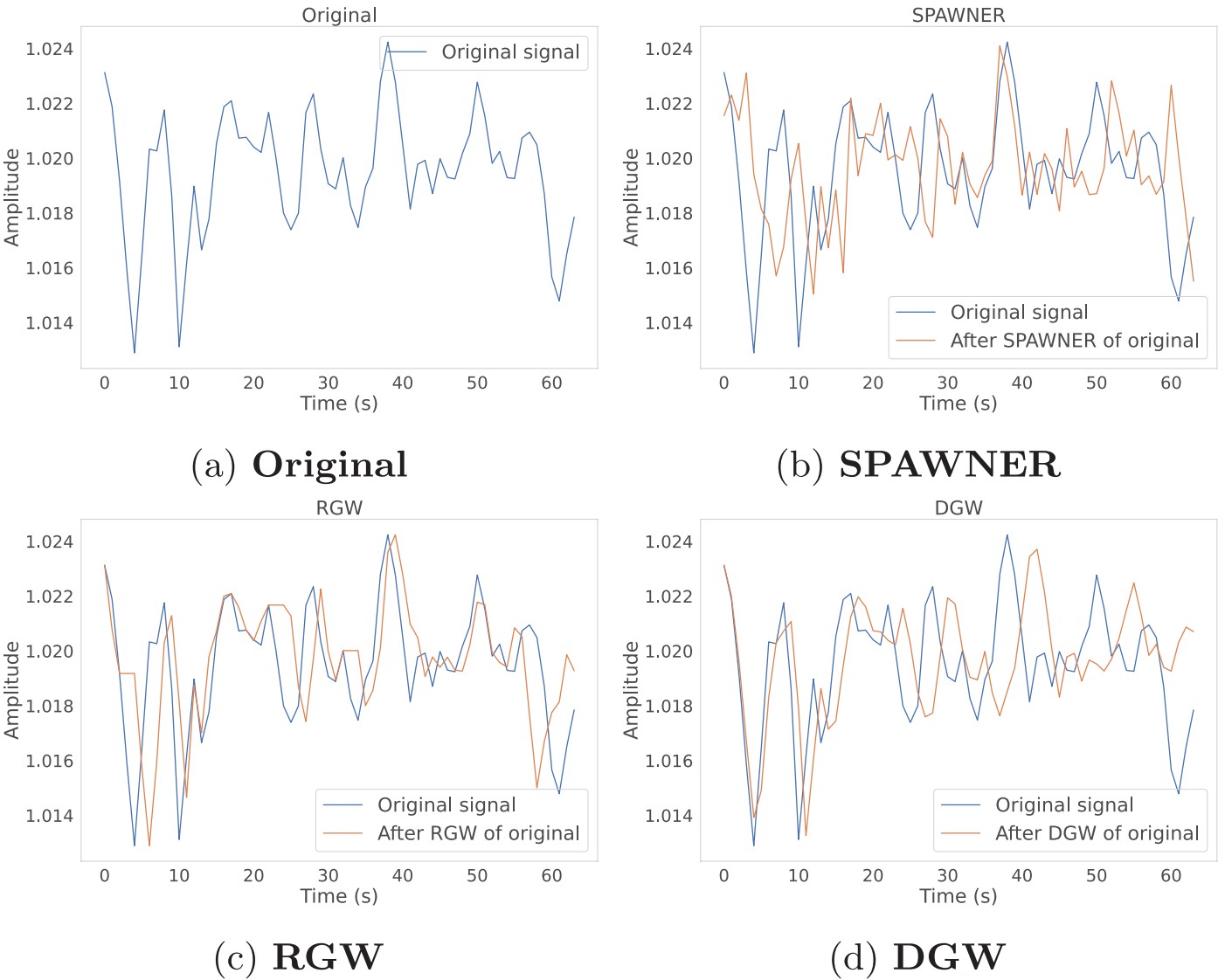

**Fig 3. Example instances of PM methods applied to a sensor channel of HAR dataset.** PM approaches hinge on the idea of aligning the time series to be augmented to another reference time series using DTW or variations of it. The main differences between the three approaches reside in the strategy used to select the reference, or the variant of DTW that they employ.

The effectiveness of SMOTE is presented in many TS applications involving gene sequence [62] and sensor data [63]. In the context of a multivariate TS dataset however, employing SMOTE to address class imbalances may inadvertently change the distribution of raw data and may produce nonsensical patterns [34].

**3.3.2 Time domain PM** In our study, we investigate the following three time domain PM approaches that were predominantly used in the literature: RGW, DGW and Suboptimal Warped Time-series Generator (SPAWNER). A description of each method is provided as follows:

**RGW** [32] is an augmentation approach that hinges on the ability of DTW to align two TS by determining a mapping between their respective elements. For each sample TS to be augmented, another sample - also referred to as the *reference* - is randomly selected from the

subset of samples sharing the same class label as the TS to be augmented. DTW is computed between the sample and its reference to obtain an augmented sequence with the values of the sample, but the timestamps of the reference.

**DGW** is a variant of RGW that incorporates two modifications proposed by Iwana et al. [39] to enhance it. Firstly, the sample and reference sequences in DGW are aligned using ShapeDTW instead of DTW, with ShapeDTW being a variant of DTW that aligns subsequences of the TS instead of the timestamp-wise elements. Secondly, DGW proposes to improve the random selection of the reference by first taking a random subset of the dataset. Sequences from this subset are then split between a positive and negative groups, depending on whether they have the same label as the sample to be augmented or not, respectively. The sequence from the subset that maximises the ShapeDTW distance between positive and negative centroids is finally chosen as reference.

**SPAWNER** [64] generates augmented samples by applying a variant of DTW referred to as suboptimal DTW to two TS selected at random from the dataset. Suboptimal DTW forces the optimal alignment found by DTW to include a specific point determined at random, i.e. it forces a specific matching between two timestamp-wise elements of each sequence.

## 3.4 Generative model

One of the main families of TSDA methods is generative algorithms. Unlike the other methods previously presented, they use generative models to create new instances of training data. During the training phase, the model learns the underlying data distribution and generates new instances based on this understanding. This approach is particularly useful when the goal is to produce diverse, realistic synthetic data that accurately Many different neural-network-based GM exist, but the most popular ones are GAN. Similar to encoder-decoder networks VAE, GAN originally introduced by Ian Goodfellow in 2014 [65] are a class of generative networks that jointly optimize two NNs referred to as discriminator (D) and a generator (G). One advantage of the GAN architecture is that it does not require labeled data to be trained. In the GAN framework, the G is in charge of creating the synthetic samples of the data distribution, while the D works to separate the actual samples from the fake ones. In this context, when D distinguishes between the two distributions, it feedbacks G negatively. In contrast, when D is unable to distinguish between the distributions, it feedbacks G positively, leading to G evolving to deceive D. In addition, D receives positive reinforcement when discrimination is performed correctly. The optimisation problem illustrated in Eq 10 describes the GAN training procedure:

$$\min_{G} \max_{D} V(D,G) = \mathbb{E}_{x \sim p_{data}(x)} \big[ log D(x) \big] + \mathbb{E}_{z \sim p_z(z)} \big[ log(1 - D(G(z))) \big] \tag{10}$$

where $V$ denotes the objective value function, and $P_{data}$ represents the probability distribution of the original data. Concurrently, $p_z(z)$ signifies the prior distribution associated with the input noise z that is sent to G. $D(x)$ denotes the probability that the sample x comes from the original data and is not produced by G.

In principle, it is anticipated that the model losses of both the G and the D should ideally be equal to 0.5 [65]. Any deviation from this equilibrium may result in GAN failure modes such as mode collapse or convergence failure. The occurrence of any of these failure modes can cause the generation of undesirable, low-quality data by the GAN.

The conventional GAN architecture is incapable of generating specific labeled data. However, with cGAN, it becomes feasible to condition the network with Supporting information such as class labels $y$ to generate specific labeled data. For this reason, we decided to include

cGAN in our comparative study. The modified training objective of cGAN is shown in Eq 11:

$$\min_{G} \max_{D} V(D, G) = \mathbb{E}_{x \sim p_{data(x)}} \left[ logD(x|y) \right] + \mathbb{E}_{z \sim p_{z(z)}} \left[ log(1 - D(G(z|y))) \right] \qquad (11)$$

## 4 Experimental comparison of TSDA methods

This section describes the comparative study of the TSDA methods tested in this study, and more specifically the five datasets related to medical wearable computing that were chosen, the baseline models that were used on each of them, and the augmentation strategy used on each dataset. Our primary objective is to conduct a comparative analysis of data augmentation techniques, emphasizing the ultimate performance of the model. We place our focus on the conclusive results rather than delving extensively into pre-processing strategies.

### 4.1 Selected datasets and baseline models

**4.1.1 Human activity recognition dataset: OPPORTUNITY and HAR** The OPPORTUNITY dataset is one of the most widely used benchmark dataset for wearable-based human activity recognition. Originally introduced for the "OPPORTUNITY Activity Recognition Challenge" by Roggen et al. [66], this dataset contains readings from motion sensors capturing various Activities of Daily Living (ADL) and has been used in many studies aiming at monitoring elderly and frail people in a home environment, such as [67,68]. The dataset includes data from various body-worn sensors that comprise 7 inertial measurement units, 12 3D acceleration sensors, and four 3D localization sensors.

Recordings involve four subjects, each contributing six runs. Five of these runs simulate ADL, representing the natural execution of daily activities, while the sixth run is a scripted "drill" session. The ADL run covers scenarios like getting up from a deckchair, grooming, relaxing outdoors, preparing and consuming coffee and sandwiches, cleaning up, and taking breaks. The drill run involves repeating a sequence of activities such as opening/closing doors and drawers, toggling lights, and performing specific actions while drinking. Annotations classify activities at different levels, including modes of locomotion, low-level actions related to objects, mid-level gestures, and high-level activity classes. The scenarios are performed in a studio flat, fostering a realistic environment for activity recognition. The dataset design encourages the natural execution of activities with user interpretation, capturing variations and nuances in daily routines. In our study, we used the 17 mid-level gestures as labels for the classification problem.

We selected the Deep convolutional and LSTM recurrent neural networks (DeepConvLSTM) model proposed by Ordonez *et. al* [4] as the benchmark model for our study as shown in Fig 4. The train, test, and validation sets are the same as the ones used by the authors. Specifically, data originating from the first and second runs, as well as the drill runs, of subjects two and three, along with all runs of subject one, are allocated to the training set. The validation set comprises data from run three of subjects two and three, while the test set incorporates data from runs four and five of subjects two and three.

In addition to the OPPORTUNITY dataset, we used the HAR dataset introduced by Anguita et al. [69] to enhance our analysis. The HAR dataset is another benchmark for human activity recognition, featuring time-series data collected from 30 participants wearing smartphones on their waists. This dataset includes accelerometer and gyroscope measurements, annotated with labels for six distinct activities: walking, walking upstairs, walking downstairs, sitting, standing, and laying. We used the same DeepConvLSTM [4] model for the classification of activities on this dataset.

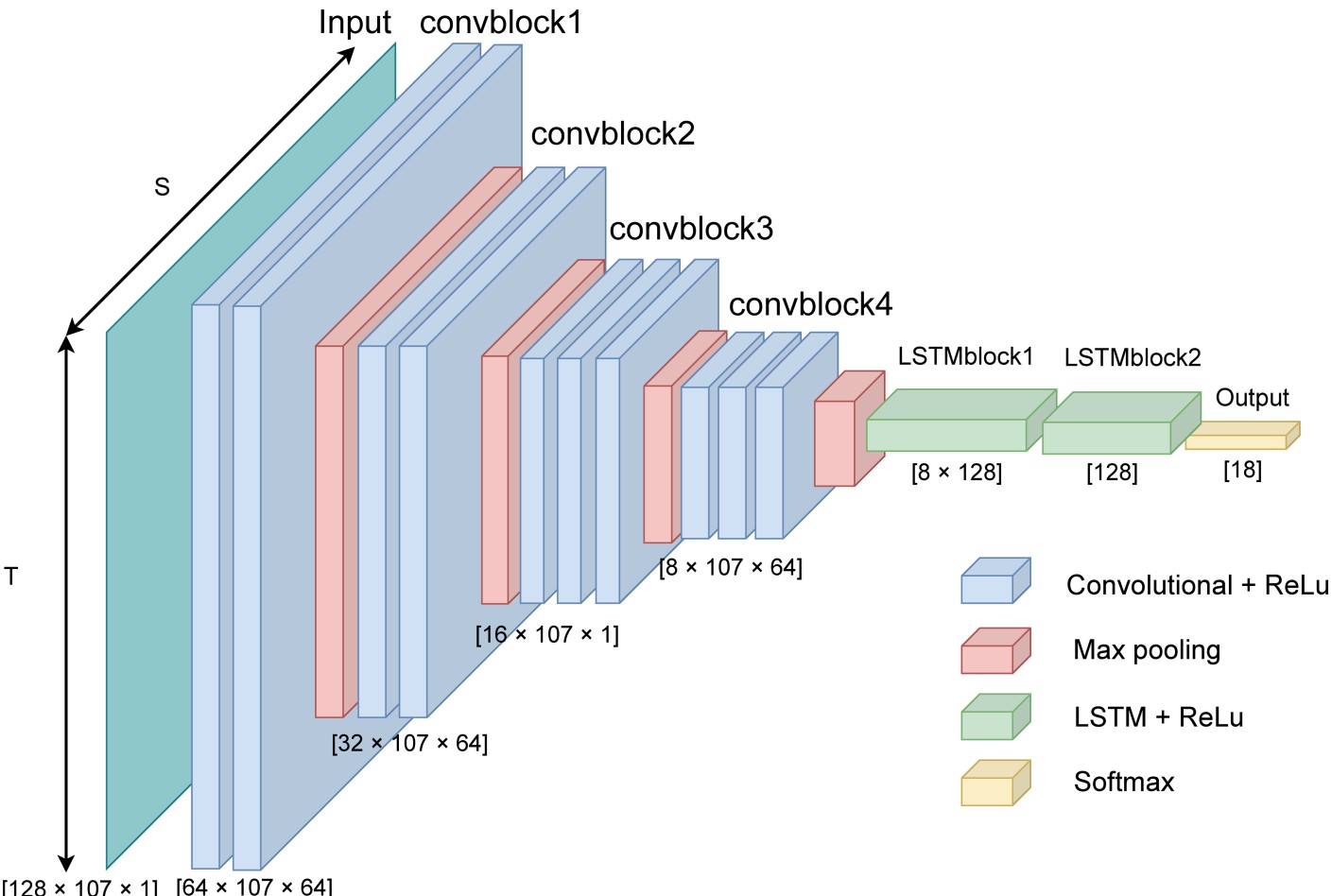

**Fig 4. Schematic illustration of DeepConvLSTM [4] architecture with four 2D convolutional, two Long Short-Term Memory (LSTM) layers, and output classes presented by a softmax layer for human activity recognition.** Initially, the different sensor channels are flattened into a (T × S)-dimensional vector and then fed to the various hidden layers.

**4.1.2 Emotion recognition dataset: DEAP** The DEAP dataset is one of the main benchmark datasets for wearable-based emotion recognition [70]. 32 subjects (16 males and 16 females aged between 19 and 37 years, with an average age of 26.9 years) participated in the data collection process. Each participant was asked to watch 40 video clips. Each video clip contained a one-minute (63 seconds, to be more specific) audio-visual music extract specifically chosen to elicit various emotional states. Physiological signals comprising Electroencephalogram (EEG) channels (n = 32) and peripheral channels (n = 8) were recorded while the participants watched the videos. The peripheral channels include Electrooculogram (EOG), Electromyogram (EMG), of Zygomaticus, Trapezius muscles, galvanic skin response (GSR), respiration amplitude, blood volume by plethysmograph, and skin temperature. In addition, before the actual recording, baseline signals were recorded for two minutes from each subject while they relaxed and looked at a fixed cross on a screen. After watching them, each participant had to rate each video regarding arousal, valence, dominance, and liking. The level of arousal, valence, dominance, and liking was assessed using the *Self-Assessment Manikin Scale* (SAM) [71], yielding numerical values between 1 (very low) and 9

(very high) for each emotional dimension. In this study, we used the pre-processed version of the data sampled at 128 Hz made available online by the authors of the DEAP dataset (https://www.eecs.qmul.ac.uk/mmv/datasets/deap/, last accessed: 08.11.2024). The dataset is available upon request (signed EULA) for research purposes. Following the common practice in the wearable-based emotion recognition literature, we focus on the two binary classification problems of low vs high arousal (level of physiological excitement and low vs high valence (level of pleasantness) in our study.

We selected the Deep convolutional 2D neural networks (DeepConv2D) model proposed by Tripathi et al. [72] as the baseline model for the DEAP dataset. It contains three 2D convolutional blocks for feature extraction, followed by a classification MLP, as shown in Fig 5. Tripathi et al. employed a window size of 310 samples per segment, conducting statistical feature engineering to generate 101 new features. Consequently, Tripathi's dataset as a dimension of $40 \times 101 = 4040$ . To reduce the volume of the preprocessed data, they applied a statistical dimensionality reduction technique. We found out that using a 2D CNN directly on the raw data (i.e. without the extraction of statistical features) led to superior results compared to the findings of Tripathi et al. We therefore use this approach as our baseline on the DEAP dataset. We utilized a 1-second sliding window size for segmentation and employed 40 channels without feature engineering.

**4.1.3 Pain recognition datasets: BVDB and PMDB**  Pain analysis and assessment is a challenging problem because pain is subjective, with a large variance between individuals, and involves emotional components. The evolutionary role of pain as a protective mechanism further complicates the assessment. The lack of a clear definition of pain and substantial progress in assessment methods make the analysis more complex. Addressing these challenges is crucial for enhancing the effectiveness of pain assessment strategies in academic research and clinical practice. To provide a comprehensive understanding of pain analysis and address the aforementioned challenges, our study incorporates two datasets, namely BioVid Heat Pain Data Base (BVDB) [73] and the PainMonit Database (PMDB) [74].

The BVDB, introduced by Walter et al. in 2013 [73], stands as the main benchmark dataset for heat-induced pain classification. The dataset integrates physiological sensors and video recordings, focusing on pain analysis. A total of 90 subjects (after data preprocessing, not all subjects are used in the training and test sets), evenly distributed across age groups (18–35, 36–50, and 51–65), participated, ensuring gender balance within each age group. The data were acquired in two phases. First, a calibration phase was performed to determine subjective pain thresholds. Then pain stimuli were induced during a stimulation phase.

In the calibration phase, pain (TP) and pain tolerance thresholds (TT) were determined, defining the range [TP, TT]. TP and TT represent the temperatures at which transitions occur from the sensation of warmth to discomfort and from bearable to unbearable pain for each individual, respectively. This range was further divided into four evenly spaced sub-intervals $[T_i, T_{i+1}]$ for $i \in \{1, 2, 3\}$, each representing a temperature level Ti during the stimulation phase. Pain stimuli of varying intensity were applied using the predefined temperature values, with each temperature administered 20 times for 4 seconds, interspersed with randomized 8–12-second pauses. The BVDB is divided into two parts (A and B) based on different sensor setups during a second pain stimulation phase. Segment A, being the more commonly referenced segment, includes video recordings, Skin Conductance Level (SCL), Electrocardiogram (ECG), and EMG data. Segment B introduces EMG data for the *corrugator*, *trapezius*, and *zygomaticus* muscles but omits video sources due to wire occlusions. For data acquisition, a comprehensive setup utilized three AVT Pike F145C cameras and one Kinect Sensor. Physiological responses were recorded using a Nexus-32 amplifier, featuring Electrodermal Activity

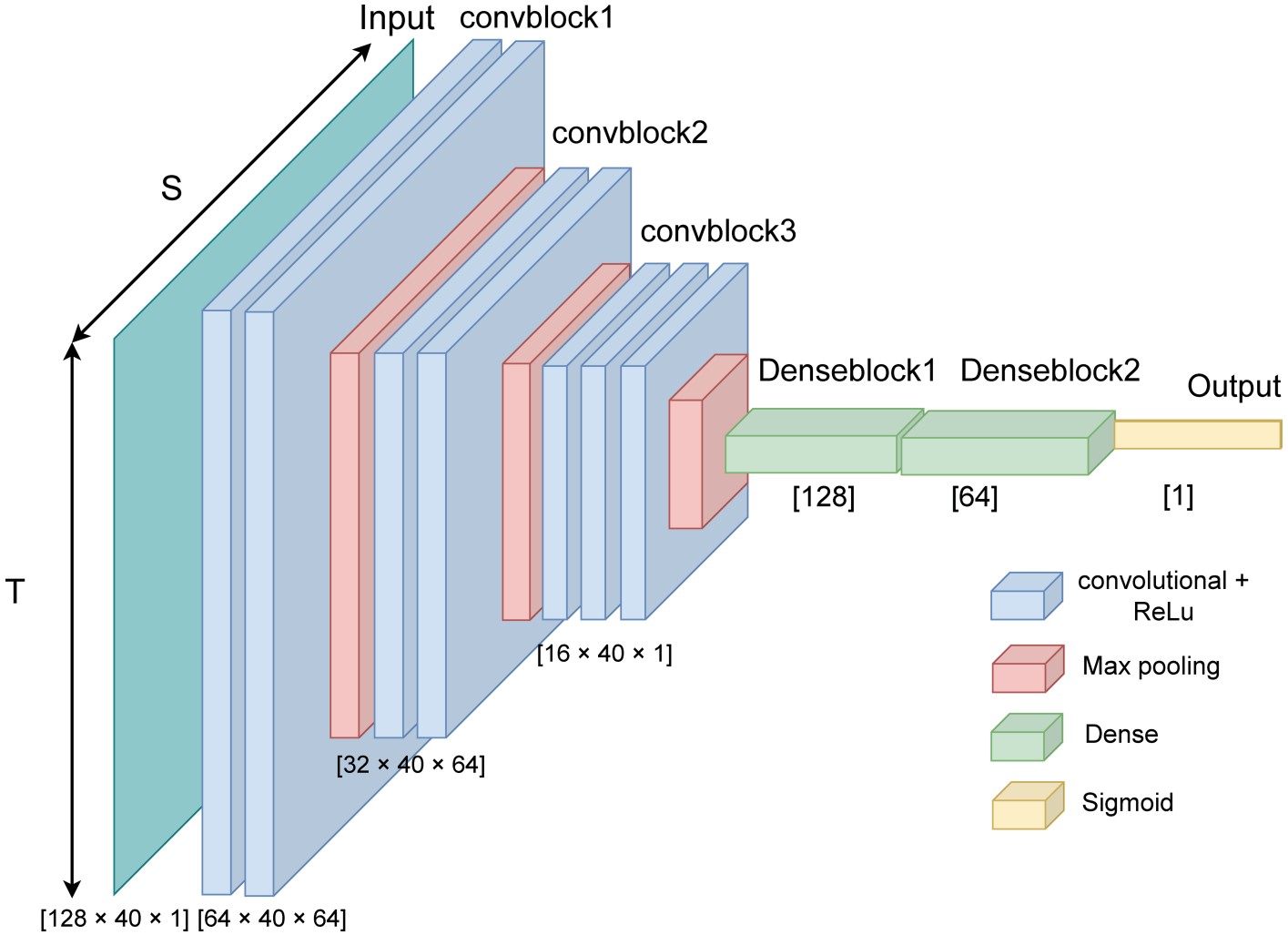

**Fig 5. Architecture of the DeepConv2D [72] framework for Human Emotion recognition.** The signals coming from the wearable sensors (128 × 40) are processed by three 2D convolutional layers, which allow learning features from the data. Two dense layers then perform a non-linear transformation, which yields the classification outcome with a sigmoid logistic regression output layer for binary label classification.

(EDA) to measure skin conductance level, ECG for heart rate, and EMG for muscle activity at three sites (*Corrugator, Zygomaticus, Trapezius*).

The PMDB, introduced by Gouverneur et al. in 2024 [74], is a dataset acquired at the Institute of Medical Informatics, University of Lübeck, Germany, focusing on heat-induced pain in a study involving 55 participants (21 male, 33 female, average age 27.35 ± 6.88). The dataset includes both objective temperature-based measurements and subjective pain annotations obtained through a computerized Visual Analogue Scale (CoVAS) slider. Participants went through a calibration phase to determine TP and TT using a thermode for pain induction. The calibration involved a staircase method with temperature stimuli ranging from 40°C to 49°C. The induction phase included eight times 10-second stimuli at specified temperatures, with participants continuously rating their pain using the CoVAS. Physiological data, including Blood Volume Pulse (BVP), Heart Rate (HR), Inter-Beats-Interval (IBI), EDA, Accelerometer (ACC), skin temperature, respiration, surface Electromyogram (sEMG), and

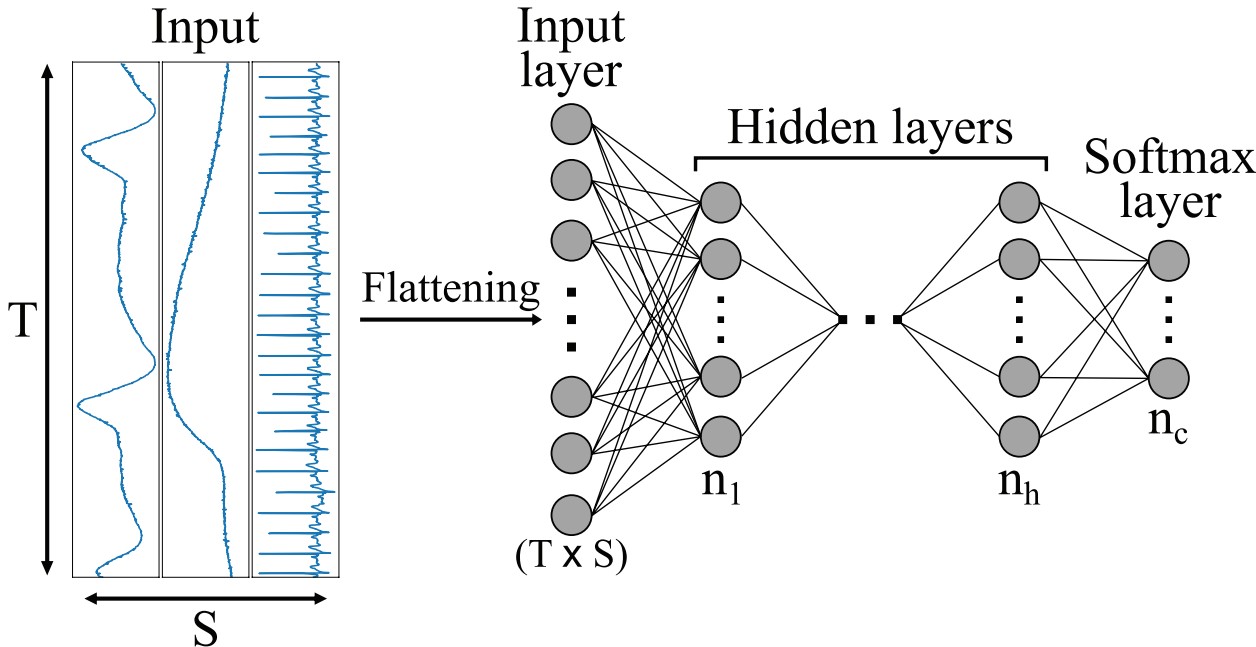

**Fig 6. Architecture of MLP model for human pain recognition.** The input is flattened first and fed into two consecutive dense layers [5].

Electrocardiography (ECG), were recorded using wearable devices. The study took approximately one hour per participant, and due to technical issues or flawed experimental setups, three subjects were removed, resulting in a final dataset of 52 subjects. The dataset offers a unique combination of subjective and objective pain measurements.

In our study, we selected no pain (TP) vs. high pain (TT) for binary label classification both for BVDB and PMDB as it is the most commonly applied task in automated pain recognition. The Multi-Layer Perceptron (MLP) proposed by Gouverneur et al. [5] is chosen as the reference model in this study to compare the efficacy of DA on both datasets. The authors claim to have the best *macro* F1 score for BVDB is (84.01%) and PMDB is 87.41% by the proposed MLP architecture shown in Fig 6. Inspired by their approach, we replicated their strategy and achieved average F1 scores of 83.88% for the BVDB and 87.12% for the PMDB dataset.

## 4.2 Pre-processing

In general, the performance of ML algorithms depends on the data quality. In cases of insufficient, unnecessary, and irrelevant data, they may provide inaccurate and less understandable results. Therefore, data preprocessing is an essential step in the ML pipeline. Since the main goal of this paper is to prove the effectiveness of DA on selected multivariate datasets, we chose state-of-the-art ML models from previous research as the benchmark for each dataset for comparison. We also keep the same data preprocessing strategy mentioned in each benchmark study. The details of the selected models on the benchmark study and their properties after preprocessing are presented in Table 1. After preprocessing the data according to the selected model strategies, the dimensions of the datasets are presented in Table 2.

**OPPORTUNITY**: The sampling frequency of the OPPORTUNITY dataset is 32 Hz, and the data were segmented with a sliding time window approach with one-second windows

**Table 1. Selected datasets properties after preprocessing.**

| Dataset | Dataset Type | sampling frequency | Window Segment | Nr. of Labels | Class distribution |
|---------|--------------|--------------------|----------------|---------------|--------------------|
| OPPORTUNITY | Multivariate | 32Hz | 1 Sec | 17 | Imbalanced |
| HAR | Multivariate | 50Hz | 2.5 Sec | 6 | Balanced |
| DEAP | Multivariate | 128Hz | 1 Sec | 2 | Balanced |
| BVDB | Univariate | 256Hz | 5.5 Sec | 2 | Balanced |
| PMDB | Univariate | 256Hz | 10 Sec | 2 | Imbalanced |

**Table 2. Properties of the selected baseline models. All models were trained five times, and their average performances are reported.**

| Dataset ($N$, $T$, $S$) | Model | Model Evaluated by | Baseline $F1$ Score (%) |
|--------------------------|-------|--------------------|--------------------------|
| OPPORTUNITY(31141, 32, 107) | DeepConvLSTM [4] | Fixed split | 58.18 |
| HAR (10299, 128, 9) | DeepConvLSTM [4] | Fixed split | 91.76 |
| DEAP(80640, 128, 40) | DeepConv2D [72] | Stratified 5-fold | 87.05 (Valence), 87.57(Arousal) |
| PMDB(1293, 2560, 1) | MLP [5] | LOSO | 87.12 |
| BVDB(3480, 1408, 1) | MLP [5] | LOSO | 83.88 |

and 50% overlap. With 107 sensor channels in total, the dimension of the OPPORTUNITY dataset is $(31141, 32, 107)$.

**HAR**: Where the HAR dataset sampling frequency and the window segment are 50 Hz and 2.5 sec. In this paper, we focus exclusively on the raw signals from nine sensor channels. This results in the dataset dimensions for both the training and test sets being $(10299, 128, 9)$.

**DEAP**: In the case of the DEAP dataset, both EEG and peripheral channels, totaling 40 sensors ($S$), were considered. Since the DEAP signals are sampled at 128 Hz and non-overlapping windows of one second were chosen, the dimensionality of the DEAP dataset in this study is $(80640, 128, 40)$.

**BVDB and PMDB**: For both BVDB and PMDB, our study focuses on using EDA signals only, as previous research has shown they are the most informative modality for heat-based pain recognition, even outperforming the combination of other sensors [5]. The raw data on both datasets were resampled to a common frequency of 256 Hz to reduce computing costs further, as suggested by the referenced paper [5]. We refined the data frames to only the EDA signal, resulting in frame dimensions of $(1293, 2560, 1)$ for PMDB with 10 sec and $(3480, 1408, 1)$ with 5.5 sec for BVDB.

The class distribution of all datasets after pre-processing is reported in Fig 7. Instead of reporting the outcome of one single-run evaluation, we decided to indicate the average results obtained after performing repeated five-fold on OPPORTUNITY and HAR, stratified five fold cross validation on DEAP, and leave-one-subject-out (LOSO) repeated five times cross-validation on BVDB and PMDB.

## 4.3 DA methods parameters

Table 3 shows the selected DA methods and their associated parameters. Each method we discussed was adopted with the same parameters from the reference papers. This way, we kept the methods to their original designs and ensured our evaluation was fair. We tested these methods by seeing how well they improved the selected models performances across various TS datasets.

**Table 3. Evaluated DA methods in this study.**

| No. | DA method | Family | Parameters |
|---|---|---|---|
| 1 | Jitter [24] | RT | $\mu = 0, \sigma = 0.03$ |
| 2 | Rotation [24] | RT | $R$ in Eq 3 |
| 3 | Scaling [24] | RT | $\alpha = 0.1$ |
| 4 | MW [24] | RT | $I = 4, \mu = 1, \sigma = 0.2$ |
| 5 | Slicing [24] | RT | crops the signal to 90% of its original length |
| 6 | TW [24] | RT | $I = 4, \mu = 1, \omega = 0.2$ |
| 7 | WW [24] | RT | $\beta \in \{0.5, 2\}$ |
| 8 | Permutation [24] | RT | Nr. segments of fixed sliced window = 5 |
| 9 | RGW [39] | PM | slope constraint = symmetric |
| 10 | DGW [39] | PM | slope constraint = symmetric |
| 11 | SPAWNER [64] | PM | slope constraint = symmetric, $\sigma = 0, \mu = 0.5$ |
| 12 | cGAN [75] | GM | See S1–S8 Tables. |

## 4.4 Augmentation strategy

In the course of our experimentation, we augmented each training set using all approaches shown in Table 3. The process involved choosing an augmentation factor $n \in ]0, +\infty[$ that determines the number of synthetic examples added to the training set as $n \times m$, where $m$ is the training set size. We used two different strategies to select the value of $n$, depending on whether the class distribution of the dataset to augment is balanced or not.

In the case of balanced datasets (HAR, DEAP and BVDB), the augmentation factor was chosen such as $n \in \{0.2, 0.4, 0.6, \ldots, 0.8, 1, 2, 3, 4\}$. Values between $n = 0.2$ and 1 provide a controlled framework for assessing the impact of augmented data volume on model training dynamics. Additionally, the value $n \in \{2, 3, 4\}$ allows us to assess the impact of performing augmentation when the augmented data volume is superior to the original training set volume. The baseline models shown in Table 2 were then trained on the augmented training set, and evaluated on real data only.

The augmentation strategy of the imbalanced datasets (PMDB and OPPORTUNITY) differs from the balanced one. We augmented only the data from the minority classes. More specifically, we tested augmentation factors ranging between 0.2 and $n_{max} \in \mathbb{N}^*$, with $n_{max}$ chosen so that $n_{max} \times N_{minority}$ is as close as possible to $N_{majority}$ without surpassing it, where $N_{minority}$ and $N_{majority}$ designate the number of samples from the minority classes and majority classes, respectively. This led us to select $n \in \{0.2, 0.4, 0.6, 0.8, 1, 2, 3, 4\}$ for the PMDB, and $n \in \{0.2, 0.4, \ldots, 1, 2, \ldots, 10\}$ for OPPORTUNITY. For each augmentation factor, the selected model underwent retraining and evaluation using the augmented dataset.

## 4.5 Evaluation

To evaluate the performances of our models, we used the several commonly used metrics for classification problems. The first of them is the accuracy, whose formula in terms of true positive (tp), true negative (tn), false positive (fp), and false negative (fn) is provided in Equation 12:

$$Accuracy = \frac{t_p + t_n}{t_p + t_n + f_p + f_n} \tag{12}$$

Among the selected datasets HAR, DEAP and BVDB are almost balanced; on the other hand, both OPPORTUNITY and PMDB are highly imbalanced (as shown in Fig 7. In this

configuration, the accuracy may be biased to reflect the performances of the dominant class only. For this reason, we computed the average $F1$ score (AF1), also referred to as the macro $F1$ score, which is the average of the $c \in \mathbb{N}^*$ classes F1 scores. The F1 score of the $i^{th}$ class $F1_i$ is defined as the harmonic mean of the considered class precision $p_i$ and recall $r_i$, whose expressions are provided in Eqs 13 and 14:

$$p = \frac{t_p}{t_p + f_p} \tag{13}$$

$$r = \frac{t_p}{t_p + f_n} \tag{14}$$

Equations 15 and 16 respectively provide the mathematical definitions of the class $F1$ score and AF1:

$$F1_i = \frac{2 \times p_i \times r_i}{p_i + r_i} \tag{15}$$

$$AF1 = \frac{1}{c} \sum_{i=1}^{c} F1_i \tag{16}$$

We also provide a comparative analysis of classification accuracy across all datasets in the appendices. It is noteworthy to emphasize that randomness introduced during the training process (e.g., random ordering of training examples, random weight initialization, etc.) makes the training of an ML model non-deterministic. This can therefore lead to some variance in the obtained results. To take this phenomenon into account, we reported the average values of each metric obtained after training the models. The outcomes of the DA methods we applied are detailed in Sect 5.

## 5 Experimental results

In this section, we present the experimental results of the tested TSDA techniques on all five benchmark datasets. All algorithms and models were implemented using Python version 3.9.17. Keras and Sklearn, with Tensorflow 2.12.0 backend libraries, were used for the deep learning architectures, respectively.

### 5.1 Comparative analysis of human activity recognition data

Because of the very notable class imbalance in the OPPORTUNITY dataset, we report the macro F1 score as the main evaluation metric. We present the comparative results in Table 4 wherein solely the $AF1$ score (%) from five repeated runs is reported for multi-class classification. According to our findings, some of the DA methods we tested led to an improvement of the model performances compared to the baseline. TW, in particular exhibits a noteworthy $F1$ score of 62.88% at an augmentation factor of 9. Our results also highlight that the cGAN performs the worst in producing meaningful full synthetic data. Although we were able to stabilize the cGAN for human activity recognition data (as shown in S1 Figure in appendices, the discriminator loss failed to converge at the ideal loss (0.5) [20].

In Table 5, we observe a notable enhancement in the AF1 score following the application of data augmentation techniques to the HAR dataset. Specifically, the baseline F1 score improved from 91.76% to 95.44% with the TW algorithm at an augmentation factor of 1. In contrast, the model performances declined when synthetic data was generated using the Rotation and GAN algorithms.

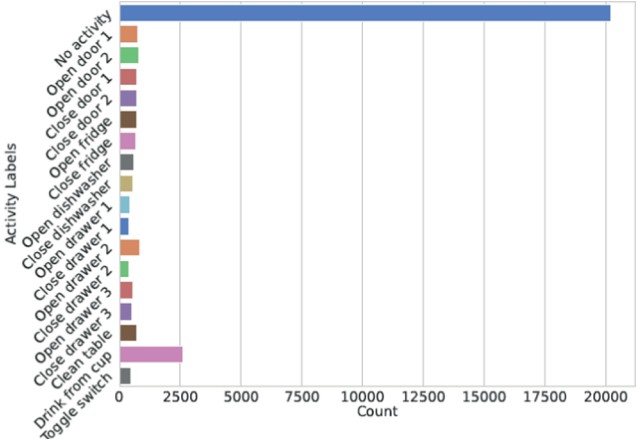

(a) OPPORTUNITY

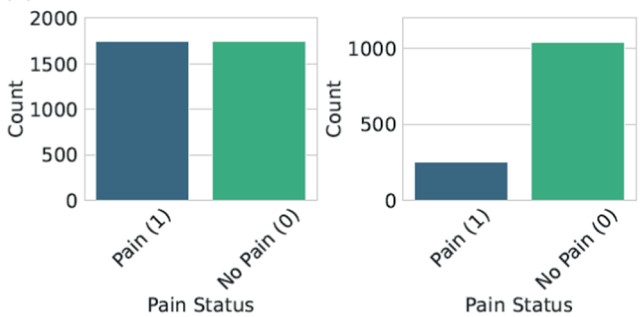

(b) BVDB (left) and PMDB (right)

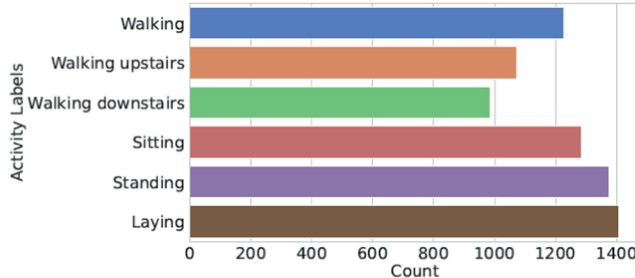

(c) HAR

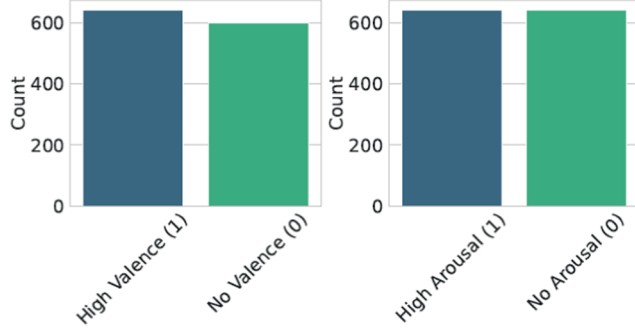

(d) DEAP

**Fig 7. Class distributions of the selected datasets.**

**Table 4. Comparative results (AF1) of DA on OPPORTUNITY. The baseline AF1 without augmentation is** 58.56%. **Lighter colors indicate better results.**

| Method / Factor | Jitter | Rotation | Scaling | MW | Slicing | TW | WW | PRM | RGW | DGW | SPAWNER | cGAN |
|---|---|---|---|---|---|---|---|---|---|---|---|---|
| 0.2 | 54.69 | 57.08 | 57.22 | 55.74 | 58.34 | 56.83 | 58.70 | 57.58 | 56.83 | 58.48 | 56.82 | 58.02 |
| 0.4 | 57.85 | 59.64 | 58.73 | 57.16 | 58.09 | 60.19 | 61.41 | 59.38 | 57.79 | 58.21 | 59.48 | 56.96 |
| 0.6 | 57.96 | 59.84 | 59.36 | 59.52 | 60.33 | 60.10 | 59.56 | 57.79 | 57.80 | 60.36 | 58.12 | 56.80 |
| 0.8 | 60.05 | 58.73 | 57.29 | 59.18 | 58.33 | 60.06 | 56.94 | 56.03 | 57.95 | 57.51 | 58.19 | 57.84 |
| 1 | 57.46 | 57.72 | 58.20 | 57.50 | 57.55 | 58.23 | 58.41 | 57.79 | 58.51 | 58.77 | 57.06 | 55.52 |
| 2 | 58.26 | 56.72 | 58.65 | 59.48 | 59.12 | 58.47 | 58.38 | 57.66 | 56.84 | 58.66 | 57.52 | 57.28 |
| 3 | 57.60 | 56.72 | 57.65 | 58.75 | 59.27 | 59.95 | 58.60 | 57.58 | 59.61 | 60.94 | 55.50 | 56.41 |
| 4 | 59.04 | 56.58 | 59.27 | 58.52 | 57.62 | 61.00 | 59.63 | 57.51 | 59.68 | 59.49 | 58.39 | 56.23 |
| 5 | 58.18 | 56.94 | 58.44 | 58.07 | 58.80 | 60.48 | 58.45 | 58.29 | 59.35 | 58.57 | 58.48 | 54.20 |
| 6 | 58.99 | 55.85 | 59.09 | 58.13 | 58.59 | 60.18 | 58.44 | 57.52 | 60.40 | 58.67 | 57.40 | 54.15 |
| 7 | 60.25 | 54.76 | 59.27 | 57.92 | 60.18 | 60.73 | 61.23 | 59.18 | 58.28 | 59.11 | 57.65 | 55.09 |
| 8 | 59.72 | 56.56 | 57.89 | 58.97 | 58.74 | 59.69 | 59.09 | 57.96 | 59.34 | 58.90 | 58.33 | 55.89 |
| 9 | 58.06 | 55.58 | 60.15 | 57.80 | 59.85 | 62.88 | 60.50 | 59.14 | 60.55 | 58.68 | 57.06 | 54.83 |
| 10 | 58.14 | 56.96 | 59.31 | 59.50 | 58.49 | 60.66 | 58.31 | 57.48 | 58.98 | 59.41 | 58.14 | 54.07 |

**Table 5. Comparative results (AF1) of DA on HAR. The baseline AF1 without augmentation is** 91.76%. **Lighter colors indicate better results.**

| Method / Factor | Jitter | Rotation | Scaling | MW | Slicing | TW | WW | PRM | RGW | DGW | SPAWNER | cGAN |
|---|---|---|---|---|---|---|---|---|---|---|---|---|
| 0.2 | 93.10 | 93.65 | 93.02 | 93.50 | 92.54 | 93.00 | 92.43 | 94.99 | 93.48 | 93.19 | 92.91 | 93.05 |
| 0.4 | 94.20 | 92.66 | 93.03 | 92.94 | 93.78 | 92.49 | 93.10 | 94.89 | 93.03 | 94.21 | 92.38 | 89.58 |
| 0.6 | 92.21 | 90.63 | 93.36 | 93.80 | 92.85 | 92.74 | 92.89 | 94.16 | 92.99 | 94.30 | 92.41 | 93.19 |
| 0.8 | 92.73 | 90.55 | 93.55 | 93.06 | 92.31 | 92.59 | 93.16 | 94.67 | 93.10 | 93.61 | 92.98 | 91.87 |
| 1 | 92.65 | 93.81 | 93.92 | 92.97 | 93.37 | 95.44 | 93.72 | 94.71 | 93.10 | 92.85 | 93.07 | 92.24 |
| 2 | 92.68 | 89.47 | 94.20 | 93.50 | 92.32 | 94.55 | 92.4 | 94.57 | 93.51 | 92.90 | 91.96 | 88.62 |
| 3 | 92.70 | 67.73 | 93.36 | 93.22 | 92.78 | 94.85 | 93.79 | 93.75 | 93.61 | 91.89 | 92.30 | 89.05 |
| 4 | 92.01 | 52.20 | 93.01 | 93.28 | 91.22 | 94.71 | 92.67 | 93.77 | 92.96 | 92.20 | 87.43 | 84.55 |

## 5.2 Comparative analysis of emotion data

We conducted an empirical investigation on the DEAP dataset, mirroring the methodology applied in our analysis of the HAR datasets. Both arousal and valence labels of the DEAP dataset are balanced (as shown in Fig 7d). Accuracy is the main evaluation metric from the literature on this dataset. Although accuracy is the primary evaluation metric used in the literature for this dataset, to keep the consistency of the comparative structure of the paper we present the results here using the F1 score. The referenced paper [72] reported that the binary label classification F1 score of valence and arousal are 87.60% and 86.49%. The effect of DA on the DEAP data towards classification is presented in Table 6 for valence and 7 for arousal. In our experimental investigation, we observed notable performance variations among selected DA methods when tuning the augmentation factor. Specifically, the TW algorithm exhibited the highest AF1 score in predicting both arousal and valence levels. A closer look at Tables 6 and 7 reveals a consistent enhancement in AF1 score across tuning factors for arousal and valence labels by the TW.

## 5.3 Comparative analysis of pain data

To check the effectiveness of DA method on human pain analysis, the BVDB and PMDB datasets went through detailed investigation in our study. In the reference paper [5], Gouverneur et al. claimed the maximum AF1 score after LOSO repeated five times is 83.88% for the BVDB dataset, and 87.12% for PMDB dataset.

**Table 6. Comparative results (AF1 score %) of DA on DEAP (Valence). Baseline AF1 score without DA is** 87.60%

| Factor / Method | Jitter | Rotation | Scaling | MW | Slicing | TW | WW | PRM | RGW | DGW | SPAWNER | cGAN |
|---|---|---|---|---|---|---|---|---|---|---|---|---|
| 0.2 | 85.30 | 85.81 | 85.17 | 85.66 | 86.12 | 85.54 | 85.98 | 85.82 | 85.10 | 85.99 | 84.72 | 85.57 |
| 0.4 | 85.26 | 84.71 | 85.26 | 85.14 | 85.71 | 86.47 | 86.15 | 85.86 | 85.65 | 85.58 | 83.85 | 85.36 |
| 0.6 | 85.55 | 84.33 | 85.28 | 85.87 | 86.31 | 86.99 | 86.41 | 86.15 | 85.96 | 85.89 | 83.55 | 84.99 |
| 0.8 | 85.69 | 83.73 | 85.65 | 85.45 | 86.45 | 87.01 | 86.43 | 86.10 | 86.17 | 86.61 | 82.37 | 85.45 |
| 1 | 85.67 | 83.31 | 85.90 | 85.94 | 86.60 | 87.39 | 86.61 | 86.37 | 86.07 | 86.69 | 82.16 | 85.21 |
| 2 | 85.78 | 81.02 | 85.72 | 85.87 | 87.77 | 88.34 | 87.31 | 87.11 | 86.44 | 87.00 | 80.73 | 85.31 |
| 3 | 85.89 | 80.35 | 85.45 | 86.02 | 87.42 | 89.17 | 87.37 | 87.59 | 86.50 | 87.00 | 79.34 | 85.11 |
| 4 | 86.01 | 79.74 | 85.80 | 86.06 | 87.53 | 89.67 | 87.79 | 88.09 | 87.00 | 87.46 | 79.60 | 83.78 |

**Table 7. Comparative results (AF1 score %) of DA on DEAP (arousal). Baseline AF1 score without DA is** 86.49%

| Factor / Method | Jitter | Rotation | Scaling | MW | Slicing | TW | WW | PRM | RGW | DGW | SPAWNER | cGAN |
|---|---|---|---|---|---|---|---|---|---|---|---|---|
| 0.2 | 86.46 | 86.02 | 86.96 | 86.88 | 86.69 | 86.99 | 86.94 | 87.02 | 86.94 | 86.86 | 86.10 | 86.72 |
| 0.4 | 86.69 | 85.59 | 86.52 | 86.69 | 87.04 | 87.69 | 87.27 | 86.91 | 86.95 | 86.81 | 85.03 | 86.70 |
| 0.6 | 86.82 | 85.29 | 86.87 | 86.71 | 87.42 | 87.78 | 87.51 | 86.95 | 86.95 | 87.11 | 84.36 | 86.42 |
| 0.8 | 86.89 | 85.27 | 86.90 | 86.86 | 87.69 | 88.45 | 87.56 | 87.21 | 87.39 | 87.42 | 83.59 | 86.28 |
| 1 | 86.93 | 83.22 | 86.53 | 87.02 | 87.67 | 88.30 | 87.76 | 87.21 | 87.64 | 87.47 | 82.76 | 86.11 |
| 2 | 86.88 | 81.29 | 86.98 | 86.93 | 88.16 | 89.67 | 87.47 | 87.80 | 87.71 | 88.12 | 81.21 | 86.41 |
| 3 | 87.30 | 78.92 | 86.92 | 87.26 | 88.43 | 90.00 | 88.34 | 88.52 | 87.79 | 88.45 | 80.22 | 85.55 |
| 4 | 86.95 | 78.20 | 87.05 | 87.45 | 89.10 | 90.54 | 88.80 | 88.39 | 88.00 | 88.50 | 79.14 | 84.76 |

The outcomes derived after applying DA are presented in Tables 8 and 9 for the BVDB and PMDB respectively. Notably, the synthetic data through the WW technique led to a noteworthy enhancement in AF1 score (84.97%), specifically a 1.10% improvement on the BVDB dataset. In a recent study by Lu et al. [76] on BVDB, an AF1 score of 85.56% was achieved for binary label classification using a sophisticated neural network framework with Multi-scale Convolutional Network, Squeeze-and-Excitation Residual Network, and Transformer encoder. Our study demonstrates that achieving a high F1 score is possible with a much simpler architecture, rather than a complex one, by incorporating a suitable data augmentation algorithm. Similarly, when considering the PMDB dataset, the $AF1$ score increased by 1.61% with the implementation of WW.

**Table 8. Comparative results (AF1 score %) of DA on BVDB. Baseline AF1 score without DA is** 83.88%

| Factor / Method | Jitter | Rotation | Scaling | MW | Slicing | TW | WW | PRM | RGW | DGW | SPAWNER | cGAN |
|---|---|---|---|---|---|---|---|---|---|---|---|---|
| 0.2 | 83.46 | 81.81 | 83.27 | 83.26 | 83.38 | 83.45 | 83.21 | 83.42 | 83.34 | 83.21 | 83.39 | 83.30 |
| 0.4 | 83.22 | 80.65 | 83.19 | 83.03 | 83.04 | 83.50 | 83.26 | 83.36 | 82.98 | 83.08 | 83.36 | 83.21 |
| 0.6 | 82.81 | 80.24 | 83.17 | 83.00 | 82.99 | 83.54 | 82.96 | 83.32 | 82.89 | 82.79 | 83.35 | 83.19 |
| 0.8 | 82.88 | 79.86 | 83.40 | 83.02 | 83.00 | 83.13 | 83.21 | 83.20 | 82.88 | 82.82 | 83.25 | 83.22 |
| 1 | 82.99 | 79.88 | 83.30 | 83.19 | 83.18 | 83.54 | 83.00 | 83.23 | 82.94 | 82.68 | 83.61 | 83.12 |
| 2 | 82.41 | 79.09 | 83.16 | 83.01 | 82.98 | 83.52 | 84.57 | 83.97 | 83.58 | 83.11 | 83.41 | 83.15 |
| 3 | 82.54 | 79.11 | 83.20 | 82.93 | 82.93 | 83.82 | 84.97 | 84.05 | 83.84 | 82.66 | 83.58 | 83.21 |
| 4 | 82.05 | 79.23 | 83.41 | 82.70 | 82.77 | 84.08 | 85.55 | 84.18 | 83.96 | 82.55 | 83.70 | 83.17 |

**Table 9. Comparative results (AF1) of DA on PMDB. Baseline AF1 score without DA is** 87.12%

| Factor \ Method | Jitter | Rotation | Scaling | MW | Slicing | TW | WW | PRM | RGW | DGW | SPAWNER | cGAN |
|---|---|---|---|---|---|---|---|---|---|---|---|---|
| 0.2 | 86.63 | 85.75 | 86.70 | 86.45 | 86.65 | 86.55 | 86.90 | 87.75 | 86.85 | 86.57 | 86.86 | 85.97 |
| 0.4 | 86.48 | 86.29 | 87.15 | 87.25 | 86.66 | 87.03 | 87.32 | 87.93 | 87.09 | 87.28 | 87.09 | 86.08 |
| 0.6 | 86.79 | 85.75 | 88.14 | 87.45 | 87.24 | 86.42 | 87.88 | 88.27 | 87.48 | 86.85 | 87.70 | 86.02 |
| 0.8 | 86.60 | 85.64 | 87.37 | 87.18 | 86.36 | 87.45 | 88.27 | 87.60 | 87.21 | 87.30 | 87.95 | 86.00 |
| 1 | 87.24 | 85.91 | 87.50 | 88.56 | 86.87 | 87.29 | 88.45 | 87.05 | 87.32 | 87.36 | 87.18 | 86.54 |
| 2 | 87.46 | 85.79 | 87.60 | 87.86 | 86.60 | 86.28 | 88.73 | 87.22 | 87.16 | 87.20 | 87.23 | 86.25 |
| 3 | 87.92 | 85.72 | 87.83 | 87.53 | 85.20 | 85.22 | 88.63 | 87.53 | 86.68 | 86.73 | 86.75 | 86.42 |
| 4 | 88.19 | 84.78 | 86.58 | 87.97 | 84.59 | 84.45 | 88.09 | 87.12 | 86.63 | 86.08 | 87.07 | 85.93 |

## 6 Statistical significance analysis of DA

To investigate the statistical significance of the previously reported results, we performed a Friedmann test to check if significant differences between the tested DA approaches (including the baseline) could be found on each dataset. The p-values of the tests are shown in Table 10. The Friedmann tests confirm that significant differences can be found on all datasets with a significance level of 0.05.

To further determine which specific methods significantly differed from one to another, we applied the Nemenyi post-hoc test using the five $AF1$ scores obtained with the best augmentation factor for each DA technique. The five $AF1$ scores obtained by the baseline (no augmentation) were also included in the test. The pairwise p-values of the Nemenyi tests carried out on each dataset are shown in Fig 8. To take into account the large family-wise error rate (FWER) caused by the high number of pairwise comparisons $c = 13 \times 12/2 = 78$, we compared the obtained p-values against the significance level adjusted by the Bonferroni correction, i.e. $0.05/c = 6.41 \times 10^{-4}$.

The analysis of the post-hoc Nemenyi test results leads to several observations. Firstly, it can be noted that there is at least one DA approach that yields significantly better performances than the baseline on all datasets. Identifying the best overall performing DA technique is not trivial because the best DA approaches vary from one dataset to another, and that several of the best ones may return comparable performances. It can nevertheless be observed that random transformation techniques overall, and more specifically TW, WW, and Permutation more consistently obtain significantly better performances than other approaches. On the opposite, cGAN often yields performances that are either not significantly different or even worse compared to the baseline.

**Table 10. Friedman test p-values obtained on each dataset.**

| Dataset | p-value |
|---|---|
| OPPORTUNITY | 1.35e−5 |
| HAR | 2.01e−5 |
| DEAP (Valence) | 2.74e−7 |
| DEAP (Arousal) | 3.84e−7 |
| BVDB | 1.50e−6 |
| PMDB | 1.35e−5 |

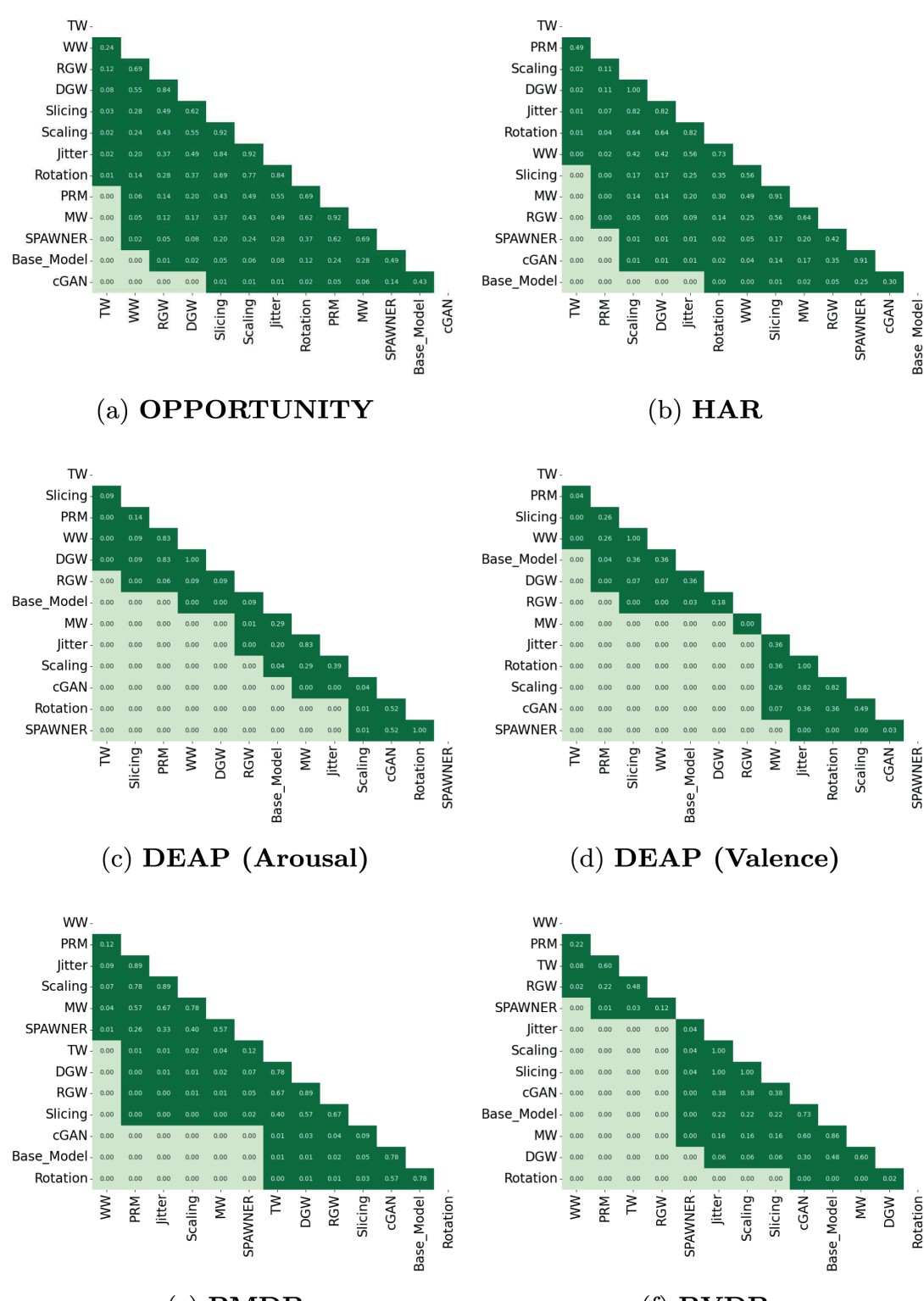

**Fig 8. p-values of the post-hoc Nemenyi tests carried out on all datasets.** All values are provided with a precision of two decimal points. DA methods and the baseline are ordered from best to worst performing according to the AF1 score on each dataset. p-values that indicate significant differences after Bonferroni correction at a significance level of 0.05 ($\leq 6.41 \times 10^{-4}$) are highlighted in light green. Non-significant differences ($> 6.41 \times 10^{-4}$) are highlighted in dark green.

# 7 Discussion

Our experimental findings further support the idea that each DA method has different effects depending on the model and dataset. Figuring out which DA method to use for TS classification in a given situation can be challenging. In this section, we provide recommendations to solve this problem based on our findings. We will answer the following two research questions based on experimental evidence and statistics for future researchers.

1. Which DA method is the most effective for TS classification?
2. How does the quantity of synthetic data impact the model performances after training?

We additionally analyse the challenges we faced in concretely applying the TSDA techniques on the datasets used in our study.

## 7.1 Optimal DA methods

In this section, we present the graphical comparison between the AF1 score of the baseline model and the AF1 score achieved by all the tested DA methods on all five tested datasets. Fig 9 compares the maximum AF1 score of selected datasets with and without DA. As depicted in Fig 9a, the time domain-based TW algorithm exhibited the highest $AF1$ score (62.88%), followed by the WW algorithm, which achieved the second-highest F1 score (61.41%) for the OPPORTUNITY dataset. It can be seen from Fig 9b that the TW exhibits the highest AF1 score (95.44%) for the HAR dataset. Simultaneously, Fig 9d and 9c illustrate the performances obtained for arousal and valence classification on the DEAP dataset. As shown in Fig 9d, the TW algorithm yielded the highest AF1 score (89.67%), with the permutation algorithm securing the second-highest AF1 score (88.09%). Regarding arousal (shown in Fig 9c), the TW algorithm achieved the highest AF1 score (90.54%), while the Slicing algorithm secured the second-highest AF1 score (89.10%). In the subsequent section, we extend our analysis regarding the results on both BVDB and PMDB datasets. Fig 9e and 9f highlight that the influence of the DA algorithm on AF1 score is minimal. The incremental improvement in AF1 score is marginal, with the WW algorithm achieving the highest AF1 score (84.98%), merely 1.1% surpassing the BVDB baseline AF1 score. Likewise, concerning the PMDB dataset, WW emerges as the most effective, attaining the highest $AF1$ score (88.73%), while MW secures the second highest AF1 score (88.56%). Our findings lead to the conclusion that, for generating more realistic synthetic pain data the WW technique proves to be more effective.

To improve the visual impact of our analysis, we present a comparative illustration between the maximum average evaluation matrices obtained by the various tested DA methods and the baseline as depicted in Fig 10. The Fig distinctly highlights that, for univariate TS datasets, the most effective augmentation algorithm is WW. This finding aligns seamlessly with the claim made by Iwana et al. in their empirical study [24]. Conversely, in the context of multivariate time series datasets, (TW) emerges as the most effective data augmentation algorithm.

Drawing conclusions from the extensive analysis, our findings underscore the nuanced nature of the impact of various augmentation factors on distinct datasets. The optimal approach varied across datasets, with the TW algorithm performing best for OPPORTU-NITY, HAR, and DEAP, and WW proving effective for BVDB and PMDB. This highlights the absence of a universal best performer that consistently achieves peak AF1 score across all TS classifications. However as shown in Table 11, it is notable that all top-performing and

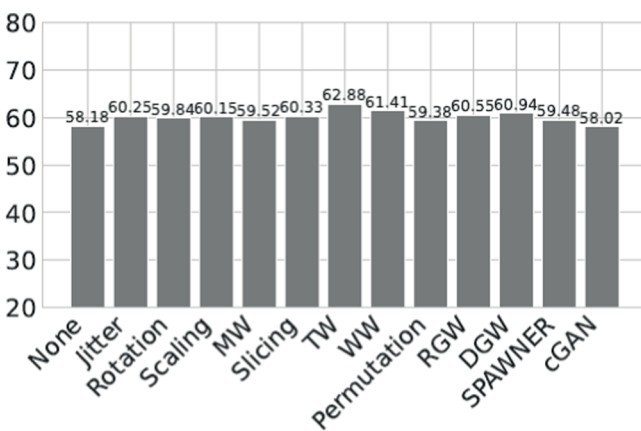

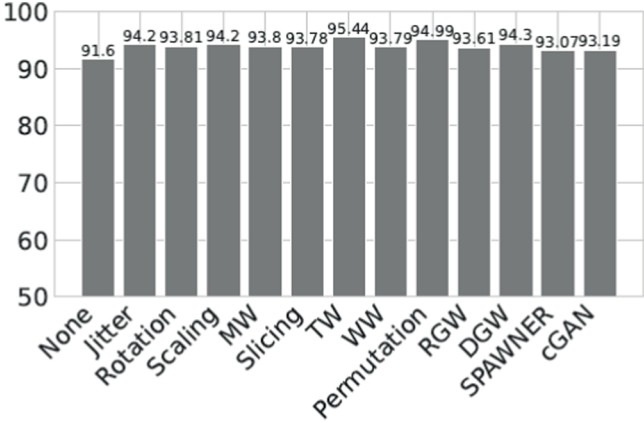

(a) AF1 score on OPPORTUNITY

(b) AF1 score on HAR

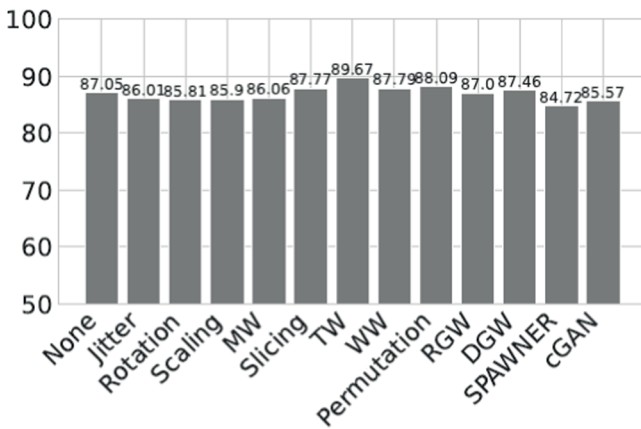

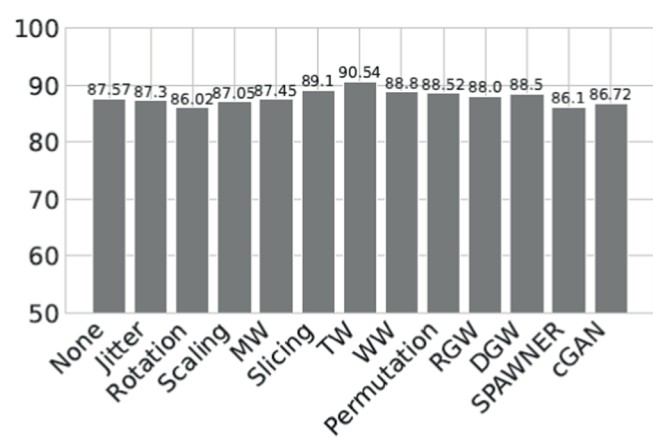

(c) AF1 score for valence on DEAP

(d) AF1 score for arousal on DEAP

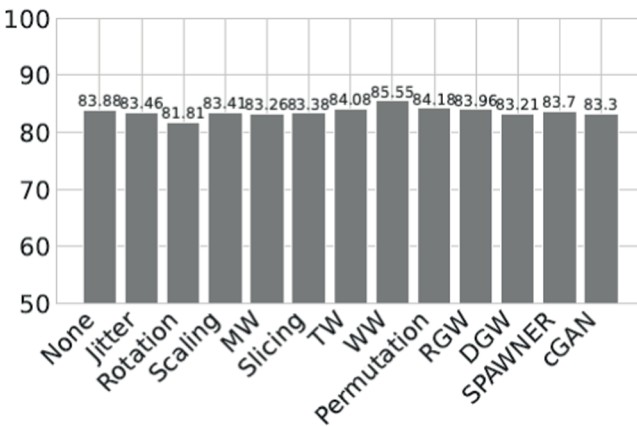

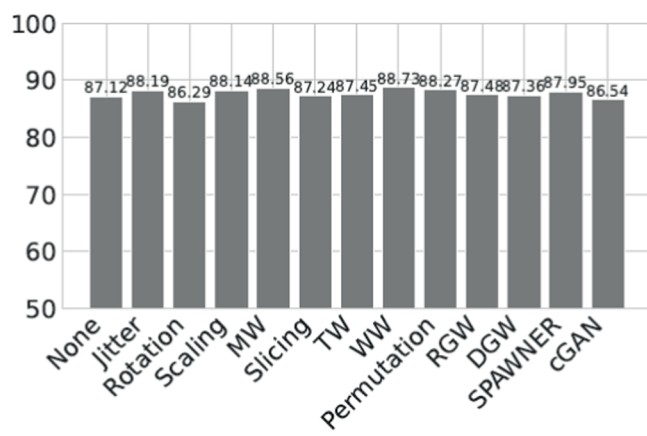

(e) AF1 score on BVDB

(f) AF1 score on PMDB

**Fig 9. Average F1 scores obtained on each dataset for all DA approaches.**

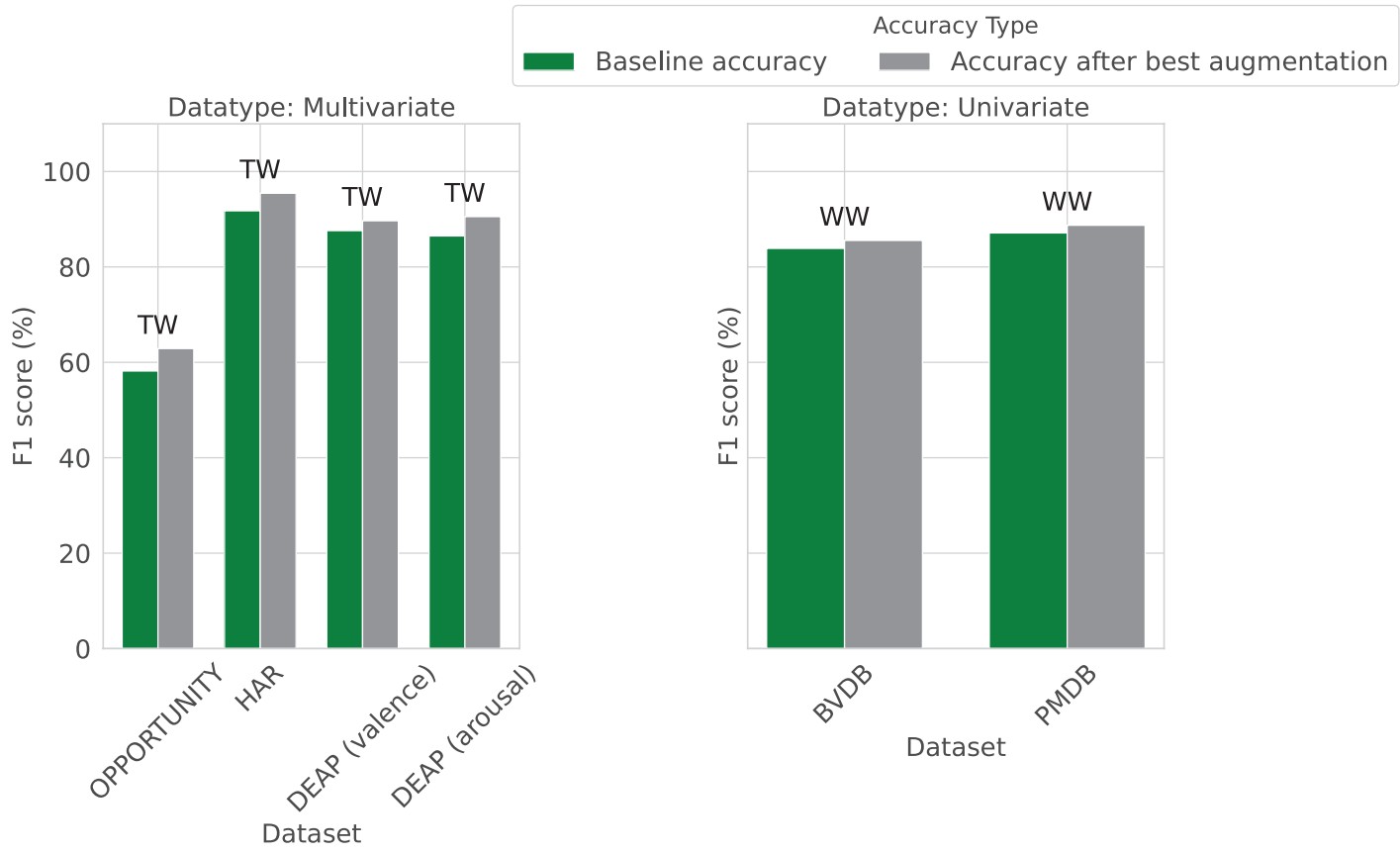

**Fig 10. Maximum AF1 score on selected datasets.**

second-best DA algorithms in our study belong to the RT family, revealing their effectiveness in enhancing TS classification outcomes. In contrast, the generative family, represented by cGAN, exhibited the poorest performance despite efforts to ensure stable training for each dataset (as shown in S1 Fig). cGAN failed to generate realistic data, marking it the least effective augmentation approach in our study.

Although the post Nemenyi analysis presented in Sect 6 shows that the top performing approaches on each dataset may yield performances that are not statistically significantly different from one another, the consistency of the obtained rankings lead us to recommend TW and WW as most effective augmentation approaches overall, as an answer to research question 1. We hypothesise that these approaches perform the best because they are the most suitable among the ones tested to deal with the general challenges of TSDA described in Sect 1.2. Since the augmentation is performed on a sample-basis (as opposed to relying on a model trained on data), these methods are more robust to the issue of data scarcity issue. Additionally, the transformations performed on the original TS by TW and WW are likely modest enough to preserve the label integrity and semantic content of the augmented data.

## 7.2 Effect of augmentation factor

To investigate the impact of the augmentation factor on model performance, we represented the AF1 score of each selected DA method across varying augmentation factors on all five

**Table 11. Rank of the evaluated TSDA methods and baseline based on their best AF1 score obtained on the five selected datasets. Approaches on each dataset are ranked from best (rank 1) to worst (rank 13).**

| DA Method / Dataset | Jitter | Rotation | Scaling | MW | Slicing | TW | WW | PRM | RGW | DGW | SPAWNER | cGAN | Baseline |
|---|---|---|---|---|---|---|---|---|---|---|---|---|---|
| OPPORTUNITY | 7 | 8 | 5 | 10 | 6 | 1 | 2 | 9 | 3 | 4 | 11 | 13 | 12 |
| HAR | 5 | 6 | 4 | 9 | 8 | 1 | 7 | 2 | 10 | 3 | 11 | 12 | 13 |
| DEAP (Arousal) | 9 | 12 | 10 | 8 | 2 | 1 | 4 | 3 | 6 | 5 | 13 | 11 | 7 |
| DEAP (Valence) | 10 | 9 | 11 | 8 | 4 | 1 | 3 | 2 | 7 | 6 | 13 | 12 | 5 |
| BVDB | 6 | 13 | 7 | 11 | 8 | 3 | 1 | 2 | 4 | 12 | 5 | 9 | 10 |
| PMDB | 3 | 13 | 4 | 5 | 10 | 7 | 1 | 2 | 9 | 8 | 6 | 11 | 12 |
| **Average Rank** | 6.67 | 10.17 | 6.83 | 8.50 | 6.33 | 2.33 | 3 | 3.33 | 6.50 | 6.33 | 9.83 | 11.33 | 9.83 |

datasets in Fig 11. Analyzing the OPPORTUNITY dataset revealed an overall increase in F1 scores with the augmentation factor, except for cGAN, Rotation, and SPAWNER (as shown in Fig 11a). A notable decline in F1 scores was evident for augmentation factors higher than 8 across all methods, hinting that adding too much synthetic data may introduce a bias in the training of the model. For the HAR dataset, the average F1 score was improved for most of the methods by an augmentation factor higher than 1, except for Rotation, GAN, and SPAWNER. For the DEAP dataset, a continuous AF1 score improvement was observed when the augmentation factor was less than or equal to 1. On the other hand, when the augmentation factor was higher than 1, indicating a higher volume of synthetic data than the original, the AF1 score decreased for all augmentation methods, as shown in Fig 11d and 11c for valence and arousal respectively. A similar trend was observed for pain datasets, particularly in the BVDB dataset, where the AF1 score declined after augmentation factor 1 for most methods, except for permutation, and WW. In the case of PMDB, the F1 score began to decrease after augmentation factor one, signifying a higher volume of generated synthetic minority data compared to the original dataset. Addressing research question 2, our analysis leads to the conclusion that

- The optimal augmentation factor depends on the dataset, classification problem, and DA method
- Overall, augmenting past a certain point stops leading to improvements, and can even cause a decline in performances. It is recommended to determine the threshold on the augmentation factor after which improvement stops being observed by performing a grid search on the augmentation parameter $n$

## 7.3 Encountered challenges

One of the main limitations highlighted by our experiments resides in the poor performances obtained by generative TSDA methods. The cGANs that were tested in this study either performed worse than the baseline not using any augmentation, or produced results not significantly different from the baseline ones on all tested datasets. These underwhelming performances are to be combined with the practical difficulties met to properly train the models, such as avoiding mode collapse where the trained model fails to produce diverse outputs. Dealing with the instability of the cGAN loss during training, which arises from the delicate balance required between the generator and discriminator, also remains challenging challenging [25,77,78]. All these factors lead us to advise against the usage of such models in their most basic form for TSDA.

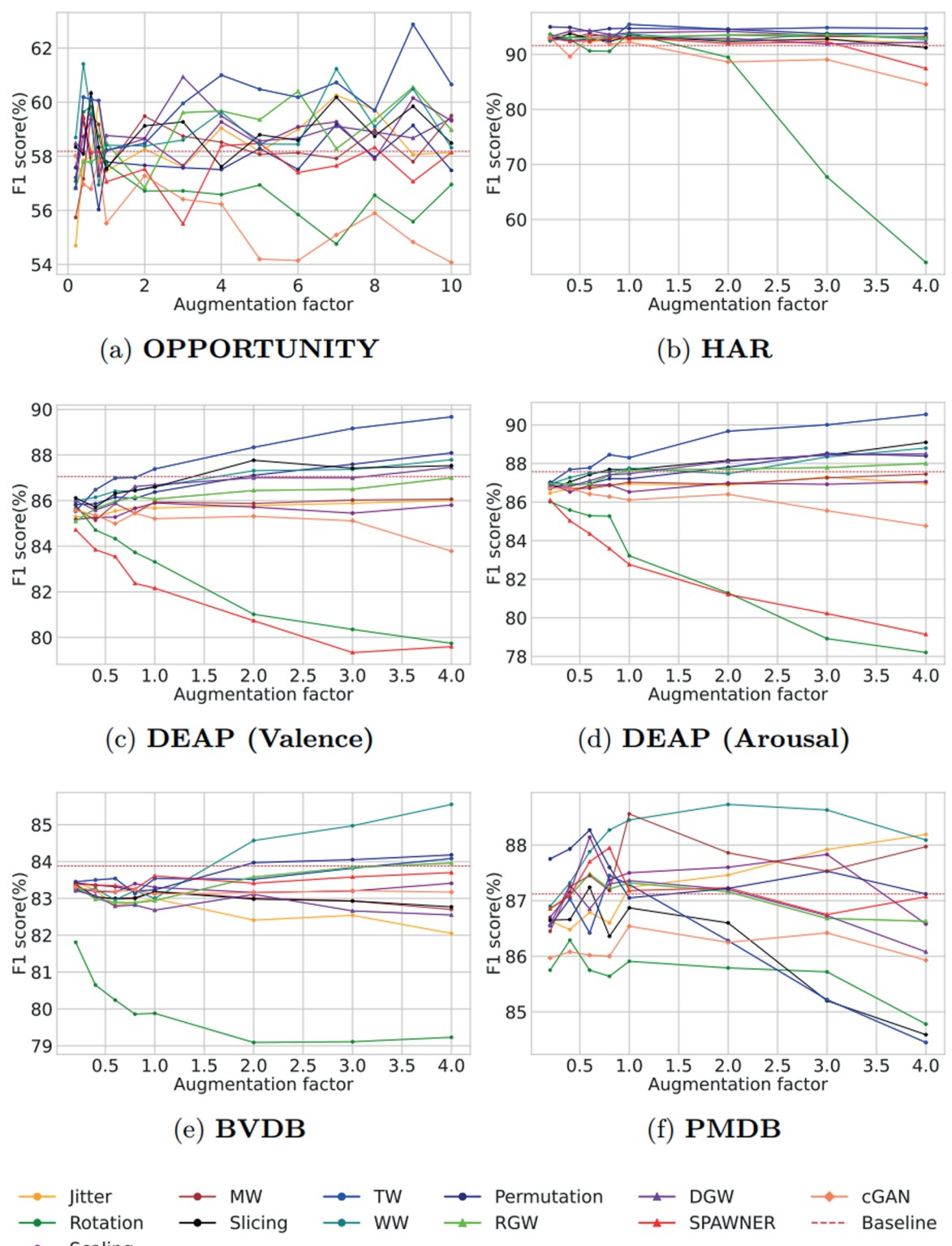

**Fig 11. AF1 scores of the DA methods in function of the augmentation factor on each dataset.** For each dataset, the horizontal dashed line represents the performance of the baseline (i.e. when no augmentation is applied).

We hypothesise that the poor performances observed by generative TSDA approaches are due to the fact that they are the most consequently affected by the challenge of data scarcity mentioned in Sect 1.2. Deep-learning-based methods are well-known to rely on large amounts of training data to learning features with a large generalisation capacity. With limited training data available, the likelihood of obtaining a model that fails to generate augmented samples that preserve label integrity or semantic content of the original data increases. Addressing these issues remains a research question that is actively investigated in the research community, mostly by experimenting with more elaborate architectures for the generator and discriminator that are more suitable for small datasets [79,80].

Finally, the statistical analysis based on the Friedmann test and post-hoc Nemenyi that we carried out would gain to be strengthened by increasing the number of runs for each DA approach. In our experiments, this number was set to five only, because of the very high number of tested configurations due to the multiple DA methods, datasets, augmentation factors and cross-validations. The optimisation of the generative models in particular incurred very significant computational costs time-wise. We believe that using a higher number of runs should be more feasible in a study focusing on a lower number of approaches and datasets.

## 8 Conclusion

In this paper, we comprehensively surveyed DA techniques applicable to both univariate and multivariate TS data and extensively evaluated them on five datasets related to wearable-based medical applications. Our study demonstrates that data augmentation (DA) techniques generally perform better than no DA across all evaluated datasets. It could also be observed that although the best performing DA approach is dataset-dependent, RT augmentation techniques perform the best overall despite their simplicity, while GM augmentation underperforms compared to other types of DA techniques. We hope this paper provides guidance to researchers and developers for selecting the suitable DA method tailored to their applications.

Despite the efforts deployed to make sure the presented experiments were carried in a thorough manner, our study still has limitations. To keep the number of tested configurations reasonable, a comparative analysis of different classification approaches on each dataset was not carried out in our experiments, and should therefore be investigated in future studies. Additionally, it would be required to check if the findings of this study generalise outside of medical-related TS datasets. Finally, despite the care taken in fine-tuning the cGAN-based models, the low performances obtained by the GM methods indicate that more investigations with generative approaches need to be performed. Future work will explore strategies consisting in optimising the generator and discriminator architectures. Another promising alternative approach consists of diffusion models that have shown significant promise in image data generation.

## Supporting information

**S1 Fig. Training loss of the Generator and Discriminator of cGAN.**
(EPS)

**S1 Table. Generator hyper-parameters.**
(PDF)

**S2 Table. Discriminator hyper-parameters.**
(PDF)

**S3 Table. Average accuracy scores of the DA approaches on OPPORTUNITY.**
(PDF)

**S4 Table. Average accuracy scores of the DA approaches on HAR.**
(PDF)

**S5 Table. Average accuracy scores of the DA approaches on DEAP for valence classification.**
(PDF)

**S6 Table. Average accuracy scores of the DA approaches on DEAP for arousal classification.**
(PDF)

**S7 Table. Accuracy scores of the DA approaches on BVDB.**
(PDF)

**S8 Table. Average accuracy scores of the DA approaches on PMDB.**
(PDF)

## Author contributions

**Conceptualization:** Md Abid Hasan.

**Formal analysis:** Md Abid Hasan, Frédéric Li.

**Funding acquisition:** Marcin Grzegorzek.

**Methodology:** Md Abid Hasan.

**Resources:** Philip Gouverneur, Artur Piet.

**Software:** Md Abid Hasan, Philip Gouverneur.

**Supervision:** Marcin Grzegorzek.

**Validation:** Frédéric Li, Artur Piet.

**Writing – original draft:** Md Abid Hasan.

**Writing – review & editing:** Md Abid Hasan, Frédéric Li, Marcin Grzegorzek.

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
