## [Decision Letter · Decision Letter 0]

14 Jun 2024

PONE-D-24-16202A comprehensive survey and comparative analysis of time series data augmentation in medical wearable computingPLOS ONE

Dear Dr. HASAN,

Thank you for submitting your manuscript to PLOS ONE. After careful consideration, we feel that it has merit but does not fully meet PLOS ONE’s publication criteria as it currently stands. Therefore, we invite you to submit a revised version of the manuscript that addresses the points raised during the review process.

We look forward to receiving your revised manuscript.

Kind regards,

Friedhelm Schwenker

Academic Editor

PLOS ONE

“This study was supported in part by the Deutscher Akademischer Austauschdienst 645

(Award No. 91831212) and within the grant SAM-SMART “Security Assistance 646

Manager for the Smart Home” (Bundesministerium für Bildung und Forschung, Grant 647

No. 16KISA074)”

3. In the online submission form you indicate that your data is not available for proprietary reasons and have provided a contact point for accessing this data. Please note that your current contact point is a co-author on this manuscript. According to our Data Policy, the contact point must not be an author on the manuscript and must be an institutional contact, ideally not an individual. Please revise your data statement to a non-author institutional point of contact, such as a data access or ethics committee, and send this to us via return email. Please also include contact information for the third party organization, and please include the full citation of where the data can be found.

4. We notice that your supplementary figure and tables are included in the manuscript file. Please remove them and upload them with the file type 'Supporting Information'. Please ensure that each Supporting Information file has a legend listed in the manuscript after the references list.

Additional Editor Comments:

Dear authors,

our peers send their comments to your manuscript. They find it interesting and suggested possible ways to improve the paper. Please take them into account when you prepare the revision.

Thank you for submitting your work to PloseOne. Best wishes!

Reviewers' comments:

Reviewer's Responses to Questions

**Comments to the Author**

1. Is the manuscript technically sound, and do the data support the conclusions?

Reviewer #1: Partly

Reviewer #2: Yes

2. Has the statistical analysis been performed appropriately and rigorously? 

Reviewer #1: No

Reviewer #2: Yes

3. Have the authors made all data underlying the findings in their manuscript fully available?

Reviewer #1: Yes

Reviewer #2: Yes

4. Is the manuscript presented in an intelligible fashion and written in standard English?

Reviewer #1: Yes

Reviewer #2: Yes

5. Review Comments to the Author

Reviewer #1: 1. The topic is interesting, however for a full-fledge survey paper I would say the coverage is not sufficient. This is rather an experimental study as they have done several experimentation to understand the impact of data augmentation.

2. The challenges in the said field should be discussed elaborately. Also discuss the motivations behind the work, shading some lights on the data augmentation aspect as one of the research challenges.

3. In the research gap section, can they be more specific on the gaps that previous surveys did not address/cover?

4. Section 2.1: They should discuss more clearly the reasons for proposing such taxonomy.

5. Can they write the captions for Figs. 2-3 in a more descriptive way to understand the differences among different data augmentation methods?

6. I suggest the authors to use more datasets.

7. I suggest the authors to use different GAN models.

8. They need to improve Section 5 significantly to understand the impact of the data augmentation issues.

9. As they are considering this as a survey paper, they need to discuss the limitations and future research avenues more strongly.

10. There are some typographical and grammatical errors. Please reread the paper and correct those.

Reviewer #2: The paper surveys data augmentation techniques for univariate and multivariate time series classification in the context of medical applications. The authors provide a comprehensive overview about data augmentation methods for such time series.

The study reports experimental results in form of mean values of 5 runs for each setup. The TSDA methods are applied to the Deap, Opportunity, Biovid and PainMonit data sets with a single learning algorithm for each dataset. However, the metrics and parameters being used are not consistent, which makes comparisons difficult. Maybe it could be helpful to provide results for all augmentation factors for all experiments (e.g. 0.2, 0.4, 0.6, 0.8 for "Opportunity" and 3-10 for the others) and F1-scores for all datasets. Moreover, it is insufficient to report mean values only, instead an appropriate measure of dispersion should be considered as well (maybe other than variance, due to the small number of samples). Something like a Critical Difference Diagram would be helpful in judging the results of the experiments (Friedman test and post-hoc Nemenyi). Simple classifiers for a baseline comparison could be helpful (something like multivariate dynamic time warping classifiers). Finally, it would be interesting to get more information about the connection between dataset characteristics and DA methods. As an example: "Rotation" impairs performance on BVDB, PMDB and Opportunity but improves performance for emotion recognition. How does this relate to the characteristics of DEAP and what general conclusion may be drawn from this?

Unfortunately, this reviewer was unable to verify the results based on the provided implementation with manageable effort (e.g. mismatching dimensions for Opportunity data).

Additional remarks:

Equations should be embedded into sentences (e.g. equation (1)).

Matrices (e.g. equation (2)) could be typeset using \vdots, \ddots, etc..

There is a typo in equation (14).

The naming of the variable "macro F_1 score" could be reconsidered. Using words in equations could be avoided.

There seems to be a mistake in the first sentence of Section 3.2.

The colors in Fig. 8 are unnecessary and might be misleading.

The y-axis in Fig. 8 could be limited to a smaller range (e.g. 50 to 90) in order to make the results more recognisable.

The notation in equation (10) and (11) could be improved.

Something seems to be missing in the second sentence of Section 6.

It could be helpful to make a difference in notation of matrices and vectors and scalars.

The parameters that appear in Table 3 are not sufficiently documented. For example \Omega is introduced as a random matrix in the context of Jittering (equation (3)). In table 3, however, it seems that \Omega denotes variance in the context of scaling. As an example, it might be more instructive to write something like $\alpha \sim \mathcal{N}(0.1,1)$ in row 3 of table 3.

The meaning of \mathbb{R}^{+*} in Section 3.4 (line 460) is unclear.

The axis labels and legends in almost all figures are not readable in a print version (Figs. 2,3,4,5,7,8,9,10,11).

The formal descriptions of DA methods in section 2 should be reconsidered. E.g.:

- It is unclear in how far equation (3) describes a rotation. The components of R hold random values between -1 and 1. The Hadamard product of this matrix and a data matrix X doesn't seem to realize a rotation.

- What is the meaning of \mathsf{T} in the exponent in equation (5)? There seems to be no need for transposition.

- equation (9) is inconsistent in notation.

6. PLOS authors have the option to publish the peer review history of their article (what does this mean?). If published, this will include your full peer review and any attached files.

Reviewer #1: No

Reviewer #2: No

---

## [Author Response · Author response to Decision Letter 1]

10 Sep 2024

We would first like to thank you for your comprehensive feedback!

Response to Reviewer 1 Comments' : We accepted all the comment and tried to answer the request in the manuscript. The details of our response is given on the "Response to Reviewers" file.

Response to Reviewer 1 Comments' : Thank you so much for your guideline to correct the mathematical equations in the manuscript. We rewrite them in the revised manuscript. We also presented a statistical analysis on our experimental observation as reviewer 2 suggested. The details of our response is presented in the "Response to Reviewers" file.

Response to Editor Comments: We want to thank the editor for guiding us through the whole process of submission. We updated our financial disclosure. Also we updated the proper contact institution for accessing the dataset used in our articles. We also response to the other comments in "Response to Reviewers" file.

---

## [Decision Letter · Decision Letter 1]

14 Oct 2024

PONE-D-24-16202R1A comprehensive survey and comparative analysis of time series data augmentation in medical wearable computingPLOS ONE

Dear Dr. HASAN,

Thank you for submitting your manuscript to PLOS ONE. After careful consideration, we feel that it has merit but does not fully meet PLOS ONE’s publication criteria as it currently stands. Therefore, we invite you to submit a revised version of the manuscript that addresses the points raised during the review process.

Please read carefully the comments of the reviewer and prepare your revision taking into account these hints.

We look forward to receiving your revised manuscript.

Kind regards,

Friedhelm Schwenker

Academic Editor

PLOS ONE

Journal Requirements:

Reviewers' comments:

Reviewer's Responses to Questions

**Comments to the Author**

1. If the authors have adequately addressed your comments raised in a previous round of review and you feel that this manuscript is now acceptable for publication, you may indicate that here to bypass the “Comments to the Author” section, enter your conflict of interest statement in the “Confidential to Editor” section, and submit your "Accept" recommendation.

Reviewer #2: All comments have been addressed

2. Is the manuscript technically sound, and do the data support the conclusions?

Reviewer #2: Yes

3. Has the statistical analysis been performed appropriately and rigorously? 

Reviewer #2: Yes

4. Have the authors made all data underlying the findings in their manuscript fully available?

Reviewer #2: Yes

5. Is the manuscript presented in an intelligible fashion and written in standard English?

Reviewer #2: Yes

6. Review Comments to the Author

Reviewer #2: This submission is an improvement over the previous one (especially Section 3). The authors have replied to all previous comments.

Parts of the statistical analysis are unclear (to me). For example: Is Figure 8 reporting on pair-wise significance tests? This would be unusual (probability for random validation grows with number of methods, see for example [Janez Demsar, 2006]).

There are still some minor issues in notation, type setting and typography. Notation and type setting are flawed especially regarding the punctuation of equations. Some captions are still not recognisable in a print version (e.g. Fig 11).

7. PLOS authors have the option to publish the peer review history of their article (what does this mean?). If published, this will include your full peer review and any attached files.

Reviewer #2: No

---

## [Author Response · Author response to Decision Letter 2]

8 Nov 2024

Please see the attached response letter.

---

## [Decision Letter · Decision Letter 2]

25 Nov 2024

A comprehensive survey and comparative analysis of time series data augmentation in medical wearable computing

PONE-D-24-16202R2

Dear Dr. Abid,

We’re pleased to inform you that your manuscript has been judged scientifically suitable for publication and will be formally accepted for publication once it meets all outstanding technical requirements.

Kind regards,

Friedhelm Schwenker

Academic Editor

PLOS ONE

Additional Editor Comments (optional):

no furhter moments

Reviewers' comments:

Reviewer's Responses to Questions

**Comments to the Author**

1. If the authors have adequately addressed your comments raised in a previous round of review and you feel that this manuscript is now acceptable for publication, you may indicate that here to bypass the “Comments to the Author” section, enter your conflict of interest statement in the “Confidential to Editor” section, and submit your "Accept" recommendation.

Reviewer #2: All comments have been addressed

2. Is the manuscript technically sound, and do the data support the conclusions?

Reviewer #2: Yes

3. Has the statistical analysis been performed appropriately and rigorously? 

Reviewer #2: Yes

4. Have the authors made all data underlying the findings in their manuscript fully available?

Reviewer #2: Yes

5. Is the manuscript presented in an intelligible fashion and written in standard English?

Reviewer #2: Yes

6. Review Comments to the Author

Reviewer #2: This reviewers comments have been addresed in a detailed fashion. The clarification of the statistical analysis in Figure 8, including the implementation of the Bonferroni correction, addresses the concerns raised effectively. Additionally, the adjustments to notation and figure readability further improve the manuscript.

7. PLOS authors have the option to publish the peer review history of their article (what does this mean?). If published, this will include your full peer review and any attached files.

Reviewer #2: No

---

## [Editor Report · Acceptance letter]

PONE-D-24-16202R2

PLOS ONE

Dear Dr. Abid,

I'm pleased to inform you that your manuscript has been deemed suitable for publication in PLOS ONE. Congratulations! Your manuscript is now being handed over to our production team.

Kind regards,

on behalf of

Prof. Dr. Friedhelm Schwenker

Academic Editor

PLOS ONE